# Regulation of Nitrate (NO_3_) Transporters and Glutamate Synthase-Encoding Genes under Drought Stress in Arabidopsis: The Regulatory Role of *AtbZIP62* Transcription Factor

**DOI:** 10.3390/plants10102149

**Published:** 2021-10-11

**Authors:** Nkulu Kabange Rolly, Byung-Wook Yun

**Affiliations:** 1Laboratory of Plant Functional Genomics, School of Applied Biosciences, Kyungpook National University, Daegu 41566, Korea; rolly.kabange@gmail.com; 2Department of Southern Area Crop Science, National Institute of Crop Science, RDA, Miryang 50424, Korea; 3National Laboratory of Seed Testing, National Seed Service, SENASEM, Ministry of Agriculture, Kinshasa 904KIN1, Democratic Republic of the Congo

**Keywords:** nitrogen use efficiency, nitrate transporters, nitrogen assimilation, *AtbZIP62* transcription factor, *AtPYD1*, drought stress, Arabidopsis

## Abstract

Nitrogen (N) is an essential macronutrient, which contributes substantially to the growth and development of plants. In the soil, nitrate (NO_3_) is the predominant form of N available to the plant and its acquisition by the plant involves several NO_3_ transporters; however, the mechanism underlying their involvement in the adaptive response under abiotic stress is poorly understood. Initially, we performed an in silico analysis to identify potential binding sites for the basic leucine zipper 62 transcription factor (*AtbZIP62* TF) in the promoter of the target genes, and constructed their protein–protein interaction networks. Rather than *AtbZIP62*, results revealed the presence of *cis*-regulatory elements specific to two other bZIP TFs, *AtbZIP18* and *69*. A recent report showed that *AtbZIP62* TF negatively regulated *AtbZIP18* and *AtbZIP69*. Therefore, we investigated the transcriptional regulation of *AtNPF6.2/NRT1.4* (low-affinity NO_3_ transporter), *AtNPF6.3/NRT1.1* (dual-affinity NO_3_ transporter), *AtNRT2.1* and *AtNRT2.2* (high-affinity NO_3_ transporters), and *AtGLU1* and *AtGLU2* (both encoding glutamate synthase) in response to drought stress in Col-0. From the perspective of exploring the transcriptional interplay of the target genes with *AtbZIP62* TF, we measured their expression by qPCR in the *atbzip62* (lacking the *AtbZIP62* gene) under the same conditions. Our recent study revealed that *AtbZIP62* TF positively regulates the expression of *AtPYD1* (*Pyrimidine 1*, a key gene of the de novo pyrimidine biosynthesis pathway know to share a common substrate with the N metabolic pathway). For this reason, we included the *atpyd1-2* mutant in the study. Our findings revealed that the expression of *AtNPF6.2/NRT1.4*, *AtNPF6.3/NRT1.1* and *AtNRT2.2* was similarly regulated in *atzbip62* and *atpyd1-2* but differentially regulated between the mutant lines and Col-0. Meanwhile, the expression pattern of *AtNRT2.1* in *atbzip62* was similar to that observed in Col-0 but was suppressed in *atpyd1-2*. The breakthrough is that *AtNRT2.2* had the highest expression level in Col-0, while being suppressed in *atbzip62* and *atpyd1-2*. Furthermore, the transcript accumulation of *AtGLU1* and *AtGLU2* showed differential regulation patterns between Col-0 and *atbzip62*, and *atpyd1-2*. Therefore, results suggest that of all tested NO_3_ transporters, *AtNRT2.2* is thought to play a preponderant role in contributing to NO_3_ transport events under the regulatory influence of *AtbZIP62* TF in response to drought stress.

## 1. Introduction

Nitrogen (N) is an essential macronutrient, which contributes substantially to the growth and development of plants [1]. The primary step of N acquisition by roots is the active transport across the plasma membrane of root epidermal and cortical cells, in the form of nitrate (NO_3_) and ammonium (NH_4_), with NO_3_ being the major source of N. The acquisition of N from the soil occurs mostly through the combined activities of low- and high-affinity NO_3_ transporters [2,3]. In higher plants, NO_3_ transporters are found within five (5) protein families, including NO_3_ transporter 1 (NRT1), NO_3_ transporter 2 (NRT2), chloride channel (CLC), and slow anion channel-associated/slow anion channel-associated homologs (SLAC/SLAH) [4,5]. In *Arabidopsis thaliana* (herein referred to as Arabidopsis), the NO_3_ transporter 1/peptide transporter (NRT1/PTR) gene family (NPF) and NRT2 have fifty-three (53) and seven (7) members, respectively. Of this number, a few genes functionally characterized were selected for this study, including *AtNPF6.2/NRT1.4* (low-affinity NO_3_ transporter [6]), *AtNPF6.3*/*NRT1.1* (dual-affinity NO_3_ transporter [7,8]), *AtNRT2.1* and *AtNRT2.2* (major components of the high-affinity uptake system) [9,10]. Members of the NRT1/PTR family (NPF) are not only involved in the uptake and transport of NO_3_ and its translocation and sensing; they also transport other biological compounds, such as peptides, amino acids, dicarboxylates, glucosinolates, indole-3-acetic acid (IAA) and abscisic acid (ABA) [11].

Nitrogen is transported from the roots to the shoot via the xylem as NO_3_, dissolved NH_3_, and amino acids. Usually [12], but not always [13], most NO_3_ reduction is carried out in the shoot, while the roots reduce only a fraction of the absorbed NO_3_ to NH_3_. After N acquisition by the roots, NO_3_ reductase (NR) and nitrite reductase (NiR) convert the exogenous NO_3_ to NH_4_, which is assimilated by glutamine synthase and glutamate synthase into amino acids [14]. Tcherkez and Hodges [15] indicated that glutamine synthase incorporates ammonia (NH_3_) as the amide group of glutamine using glutamate as a substrate in the chloroplasts. Glutamate synthase (Fd-GOGAT and NADH-GOGAT) transfers the amide group onto a 2-oxoglutarate molecule producing two glutamates. In the process, transaminations take place to make other amino acids (most commonly asparagine) from glutamine. The enzyme glutamate dehydrogenase (GDH) protects the mitochondrial functions during periods of high N metabolism and takes part in N remobilization [16]. To maintain an ionic balance, every NO_3_ molecule taken into the roots must be accompanied by either the uptake of a cation or the excretion of an anion. Plants take up metal ions like potassium (K^+^), sodium (Na^+^), calcium (Ca^2+^), and magnesium (Mg^2+^) to exactly match every NO_3_ taken up and store these as the salts of organic acids like malate and oxalate [17].

However, external fluctuations of N supply to plants caused by abiotic stress such as drought hinder NO_3_ acquisition events due to water scarcity. As a result, the reduced water availability affects the whole N use necessary for plants to complete their life cycle [18], including N uptake and transport, translocation, assimilation, and redistribution (also referred to as nitrogen use efficiency, NUE). Under the same conditions, a transcriptional reprogramming takes place, resulting in the activation or suppression of abiotic stress-responsive genes, as well as the induction of various signaling networks to optimize N uptake and utilization [19]. Ashoub, et al. [20] reported that the activity of a large number of enzymes involved in N assimilation decreases under long-term abiotic stress conditions, while Wendelboe-Nelson and Morris [21] supported that the activity of other enzymes involved in the same process increases under short-term exposure.

Protein–DNA and protein–protein interactions are fundamental to nearly all biological processes in all biological systems [22]. The joint actions of transcription factors (TFs) regulate the expression of genes in biological systems. TFs bind to *cis*-regulatory elements present in the promoter region of genes [23]. TFs operate either alone or in complex with other molecules to activate or repress the recruitment of the basal transcriptional machinery to specific genes [24], thereby determining when and where the target genes are transcribed, and the level of protein synthesis, as well as the resulting phenotype [22]. Basic leucine zipper proteins (bZIP) are transcription factors (TFs) involved in various developmental processes, including seeds maturation, plant growth, flower development, and signaling during abiotic and biotic stresses [25].

Recently, a study attempting to characterize a member of the bZIP TF family, *AtbZIP62* TF, proposed this gene as a positive regulator of drought stress response in Arabidopsis [26]. Thus, from the perspective of exploring the transcriptional regulation of NO_3_ transporters and assimilation-related genes under drought stress, we performed an in silico analysis, which helped to identify *cis*-regulatory elements specific to two members of the bZIP family (*AtbZIP18* and *AtbZIP69* earlier suggested to be regulated by *AtbZIP62* TF), in the promoter of the target genes. Similarly, *AtbZIP62* was proposed to interact with *AtbZIP18* or *AtbZIP69* TFs to regulate the expression of *AtPYD1* (*Pyrimidine 1*, a gene catalyzing the step-limiting factor of the *de novo* pyrimidine biosynthesis pathway, which in turn is proposed to have a crosstalk with the N metabolism) under drought stress [26].

Therefore, in the present study, we investigated the role of *AtbZIP62* TF in the regulation of the expression of four (4) NO_3_ transporters and two glutamate synthase-encoding genes involved in N assimilation in Arabidopsis in response to drought stress. Furthermore, we were interested to see how the proposed transcriptional interplay of *AtbZIP62* with *AtPYD1* would contribute to elucidating the mechanism underlying the regulation of NO_3_ transporters and assimilation-related genes by *AtbZIP62* TF. To achieve that, the Arabidopsis Col-0 (wild type, WT), and the *atbzip62* and *atpyd1-2* mutant lines lacking *AtbZIP62* and *AtPYD1*, respectively, were used as genetic materials.

## 2. Results

### 2.1. In Silico Transcription Factor Binding Sites Analysis Identified Cis-Regulatory Elemenents for bZIP TFs in NO_3_ Transporters and Glutamate Synthase-Encoding Genes

In our previous studies, *AtbZIP18* and *AtbZIP69* were proposed to be co-expressed, while being negatively regulated by *AtbZIP62* TF under salinity [27] and drought stress [26]. In this study, the results in Table 1 show the presence of two binding sites specific to *AtbZIP18* and *AtbZIP69* TFs in the promoter of *AtbZIP62* TF; while in that of *AtPYD1*, one binding site for *AtbZIP69* TF was found. In addition, two (2) binding sites for *AtbZIP18* and three (3) for *AtbZIP69* were identified in the promoter of *AtNPF6.2/NRT1.4*. In the same way, two binding sites for *AtbZIP18* and *AtbZIP69* were detected in the promoter of *AtNRT2.2*. Furthermore, a single binding site for *AtbZIP18* was found in the promoter of *AtGLU1*. However, no binding sites for *AtbZIP18* or *AtbZIP69* were identified in the promoter of *AtNPF6.3*, *AtNRT2.1*, and *AtGLU2*.

### 2.2. Prediction of Protein–Protein Interaction Network of Nitrate Transporters and Assimilation

To further understand the mechanism underlying the transcriptional regulation of the NO_3_ transport and assimilation-related genes, as well as their possible interaction networks, we employed the STRING database (version 11.5, https://string-db.org, accessed on 9 September 2021), which proposes protein–protein interactions in plants. Because *AtNRT2.2* (a high-affinity NO_3_ transporter) exhibited the highest transcriptional level (upregulated by 94.2-fold change) in response to drought stress, we selected the protein sequence of this gene to unveil the identity of proteins predicted to interact with NRT2.2.

At first, we were interested to see the possible functional interactome involving NRT2.2 and other NO_3_ transporters. The results of the protein–protein interaction prediction revealed that NRT2.2 has been experimentally determined to functionally interact with WR3 (AT5G0200, a high-affinity NO_3_ transporter 3.1 acting as a dual component transporter with NRT2.1, and said to be involved in targeting NRT2 proteins to the plasma membrane) (Figure 1, Table 2). What is common in the predicted interactome is that NRT2-encoding genes (NRT2.1 and NRT2.2) do not show direct functional interactions or synergy, but both are predicted to have a relationship with WR3 (would-responsive gene 3, encoding a high-affinity nitrate transporter), NIR1, NIA1 and NIA2 (Figure 1C,D). Similarly, NFP6.3/NRT1.1 is predicted to interact with NRT2.2, while NFP6.2 is proposed to be functionally associated with NRT2.2 and WR3. It was observed that NIA1 and NIA2, as well as WR3 appear to functionally interact with both dual- and high-affinity NO_3_ transporter (NRT6.3/NRT1.1, NRT2.1 and NRT2.2), except NPF6.2 (a low-affinity NO_3_ transporter) (Figure 1A–D).

In Arabidopsis, N assimilation is mediated by a variety of enzymes, including GLU1 and GLU2 known for their important roles during the early steps of the process. GLU1 is a chloroplastic/mitochondrial ferredoxin-dependent glutamate synthase 1 (Fd-GOGAT), proposed to be involved in photorespiration and N assimilation, while GLU2 is a chloroplastic ferredoxin-dependent glutamate synthase. These proteins have the ability to supply a constitutive level of glutamate to maintain a basal level of protein synthesis, and are suggested to play a role in primary N assimilation in roots. Here, the STRING results proposed that both GLU1 and GLU2 would have a functional protein–protein interaction with same protein counterparts, including two cytosolic glutamine synthase (GLN1.4, AT5G16570; GSR2, AT1G66200) and a chloroplastic/mitochondrial glutamine synthase (GS2, AT5G35630) identified as being responsible for the reassimilation of the NH_3_ generated by photorespiration. In addition, a set of three glutamate dehydrogenase-encoding genes (*AtGDH1*, AT5G18170; *AtGDH2*, AT5G07440; *AtGDH3*, AT3G03910, associated with N assimilation) were suggested (experimentally determined as indicated by the purple connecting line) to have common functional target protein interactions, with a few exceptions like NIR1 (AT2G15620, identified as a chloroplastic ferredoxin-nitrite reductase involved in the second step of NO_3_ assimilation) only linked with GLU2 (Figure 1E,F).

### 2.3. KEGG Analysis Suggests a Crosstalk between De Novo Pyrimidine Biosynthesis and the Nitrogen Metabolic Pathways

A possible crosstalk between the de novo pyrimidine biosynthesis pathway and the nitrogen metabolism was explored. In this study, we included *AtPYD1*, a gene catalyzing the step-limiting factor of the de novo pyrimidine biosynthesis pathway. As indicated in the panels A and B of Figure 2, de novo pyrimidine biosynthesis pathway is shown to have a crosstalk with the nitrogen metabolic pathway, and carbamoyl phosphate is proposed to serve as a common substrate for both pathways.

### 2.4. AtNPF6.2/NRT1.4, AtNPF6.3/NRT1.1 and AtNRT2.2 Were Similarly Regulated in atbzip62 and atpyd1-2 Mutants, While ANRT2.1 Showed Differential Expression Pattern under Drought Stress

Nitrate uptake, transport and assimilation are strongly affected when plants experience reduced water potential as well as physical water unavailability. Here, data show that the low-affinity NO_3_ transporter *ANPF6.2/NRT1.1* as well as the dual-affinity (*AtNPF6.3*/*NRT1.1*) and the high-affinity (*AtNRT2.1* and *AtNRT2.2*) NO_3_ transporter-encoding genes were significantly upregulated by drought stress in Col-0 WT (Figure 3A–D), with *AtNRT2.2* having the highest transcript accumulation level (upregulated by about a 94.2-fold change). Under the same conditions, the expression of *AtNPF6.2*/*NRT1.4*, *AtNPF6.3*/*NRT1.1*, and *AtNRT2.2* was downregulated in the *atbzip62* mutant (Figure 3A,B,D). Meanwhile, *AtNRT2.1* showed a similar transcript accumulation pattern with Col-0 (Figure 3C).

In addition, *AtNPF6.2/NRT1.4* transcript accumulation decreased significantly in *atpyd1-2* compared to that of Col-0. However, the expression of *AtNPF6.3*/*NRT1.1* was significantly downregulated in a similar manner with the one observed in *atbzip62*. Similarly, the expression of *AtNRT2.2* in *atpyd1-2* showed a downregulation pattern (Figure 3D). However, *AtNRT2.1* transcript accumulation was not affected by drought stress in *atpyd1-2* (Figure 3C). The *AtbZIP62* TF was recently proposed to be involved in the regulation of the expression events of *AtPYD1*, when investigating their transcriptional interplay under drought stress [26].

### 2.5. Drought Stress Differentially Regulated AtGLU1 and AtGLU2 Genes

Glutamate synthase enzymes are key enzymes identified as substrates for glutamine synthase in the initial steps of N assimilation in plants under optimal growth conditions. Here, our data show that the expression of *AtGLU1* and *AtGLU2*, two genes encoding glutamate synthase, was differentially regulated by drought stress, with *AtGLU1* being downregulated and *AtGLU2* upregulated in both Col-0 and *atbzip62* (Figure 3E,F). Meanwhile, the transcript accumulation of *AtGLU1* increased significantly in *atpyd1-2*. In contrast, the expression of *AtGLU2* reduced significantly in *atbzip62* and *atpyd1-2* compared to that recorded in Col-0.

Previous studies suggested that *AtbZIP18* and *AtbZIP69* would co-express under salinity [27] and drought stress [26], and *AtbZIP62* TF is identified as a negative regulator of both genes. In the same studies, the authors proposed that *AtbZIP62* TF would interact with *AtbZIP69* TF to regulate the expression of *AtPYD1*. As show in Table 1, in silico transcription factor binding sites analysis detected *cis*-regulatory elements specific to *AtbZIP69* in the proximal promoter of *AtPYD1* and that of *AtGLU1*. Therefore, we measured the expression of *AtbZIP18* and *AtbZIP69* in *atpyd1-2*. Data in the panels G and H of Figure 1 reveal that the transcript accumulation of both *AtbZIP18* and *AtbZIP69* decreased significantly in *atpyd1-2* compared to that observed in Col-0.

Because *AtNRT2.2* recorded the highest transcript accumulation (94.2-fold change) upon drought stress induction in Col-0 WT, we were interested in investigating in detail the possible interactome underlying its regulatory network. The results in Table 2 give insights into the possible functional protein–protein interactions that AtNRT2.2 (protein) may have; these include the NIR1 (a ferredoxin-nitrate reductase involved in the second step of nitrate assimilation) [28], NIA1 and NIA2 (encoding NO_3_ reductase, and involved in nitric oxide (NO) biosynthesis) [29], WR3 (known as having a high-affinity for NO_3_ transport, acting as a dual component with NRT2.1) [30], NPF6.3/NRT1.1 (a dual-affinity NO_3_ transporter involved in the regulation of NRT2.1) [31], and NRT1.2 (a low-affinity NO_3_ transporter mediating constitutive nitrate uptake) [32]. This may imply that AtNRT2.2 would play a key role during N acquisition and transport when plants experience drought stress conditions, while interacting with a set of other NO_3_ transporters and NO_3_ reductase-encoding genes.

### 2.6. Proposed Signaling Model of AtbZIP62 TF and NO_3_ Transporters and Glutamate Synthase under Drought Stress

We have summarized the data and proposed a signaling model primarily using the recorded gene expression data of the analyzed NO_3_ transporters and glutamate synthase-encoding genes in Col-0 (WT), and the *atbzip62* and *atpyd1-*2 mutant lines, in response to drought stress (Figure 4). Here, *AtbZIP62* is shown to positively regulate the expression of *AtPYD1* under drought stress, as previously suggested [26]. Then, based on the transcript accumulation patterns of *AtNPF6.2/NRT1.1*, *AtNPF6.3*/*NRT1.1* (low- and dual-affinity NO_3_ transporters), *AtNRT2.1* and *AtNRT2.2* (high-affinity NO_3_ transporters), and *AtGLU1* and *AtGLU2* (encoding glutamate synthase) in the mutants compared with the wild type, we proposed a positive or negative regulation by *AtbZIP62* TF. This signaling model also considered the known and predicted interactions between the target genes in the present study to draw a possible linkage between genes, based on protein–protein interaction by STRING and the literature. Nitrate transporters are widely known for their roles in mediating the mobility, sensing, and translocation of nitrogen (N), while the assimilation of N by plants in the form of amino acids involves a glutamine–glutamate synthase complex, interacting with other compounds. Our results give new insights into the transcriptional regulation mechanism of some of the key NO_3_ transporters, while highlighting the regulatory role of *AtbZIP62* TF and *AtPYD1* under drought stress.

## 3. Discussion

### 3.1. AtbZIP62 Regulates Nitrate Transporter-Encoding Genes in Response to Drought Stress

Studies describing the physiology and biochemistry aspects of plant nutrition have established that macronutrients such as nitrogen (N) are acquired from the soil, through the combined activities of low- and high-affinity NO_3_ transport systems across the plasma membrane of epidermal and cortical cells of roots [2,3]. In aerobic soils where nitrification can occur, NO_3_ is usually the predominant form of available N that is absorbed [35,36]. Nitrate is taken up by several NO_3_ transporters, which use a proton gradient to power the transport [37,38]. Currently, only a few NO_3_ transporters have been associated with the coping mechanism under limited N availability caused by adverse environmental conditions [31,34,39,40,41,42,43] in Arabidopsis and rice. For instance, *AtNPF7.2/NRT1.8*, a low-affinity NO_3_ transporter, is believed to be actively involved in NO_3_ uploading from xylem vessels in response to cadmium (Cd) stress [44]. Another essential component in the regulation of NO_3_ reallocation, *AtNRT1.5* (a xylem NO_3_-loading transporter) was proposed to be downregulated by salt and Cd stress in Arabidopsis [45]. The authors supported that *AtNPF7.2* regulates *AtNRT1.5*. Léran, et al. [46] reported that *AtNPF6.3/NRT1.1* is involved in NO_3_ influx and translocation to the shoot [7,47]. In the same way, *AtNPF6.3/NRT1.1* (also known as *CHL1*, *chlorate resistance 1*) was reported to act as an NO_3_ sensor [48,49,50,51], and participates in the signaling pathway triggering root colonization of NO_3_-rich patches [52].

In this study, the recorded downregulation patterns of *AtNPF6.2/NRT1.4*, *AtNPF6.3/NRT1.1* (low- and dual-affinity NO_3_ transporters), and *AtNRT2.2* (the high-affinity NO_3_ transporter) between Col-0 WT and *atbzip62* and *atpyd1-2* in response to drought stress would suggest a positive regulation by *AtbZIP62* TF and *AtPYD1*. *AtNRT2.2* exhibited the highest transcript accumulation level in response to drought stress (about a 94.2-fold change) in Col-0. Table 2 identified NRT1.5 (AT1G32450) and NRT1.7 (AT1G69870) as potential targets for *AtNRT2.2* during NO_3_ transport. Meanwhile, the expression of *AtNRT2.1* showed a similar transcript accumulation pattern with Col-0 WT (Figure 3C), which may imply that *AtbZIP62* TF may not be involved in the transcriptional regulation events of *AtNRT2.1* under drought stress. However, the significant decrease in the transcript accumulation of *AtNRT2.1* in *atpyd1*-2 would imply that *AtPYD1* might be involved in the positive regulation of *AtNRT2.1* under drought stress.

Under these conditions, *AtbZIP62* TF, known as a positive regulator of drought stress response, would play a key role in promoting NO_3_ transport activity. This event may involve *AtPYD1*, considering their regulatory records as well as the crosstalk between the de novo pyrimidine biosynthesis pathway and the nitrogen metabolism (Figure 2A,B), supported by the predicted functional protein–protein interaction between PYD1 (Pyrimidine 1) and CARB (carbamoyl phosphate) (Figure 1G). Therefore, *AtbZIP62* TF and *AtPYD1* are proposed to positively regulate the transcriptional events of NO_3_ transporter-encoding genes in response to drought stress. In a converse approach, Zhong, et al. [53] supported that a member of the bZIP transcription factor family, *AtTGA4/AtbZIP57* (AT5G10030) confers drought tolerance by promoting NO_3_ transport and assimilation.

Owing to these results, we could speculate that upon drought stress induction, the availability of NO_3_ may be affected significantly due to water scarcity. Under these conditions, *AtNRT2.2* is believed to play a preponderant role in NO_3_ transport when plants experience drought stress, under the regulatory influence of *AtbZIP62* TF, while interacting with other NO_3_ transporters as proposed in Table 2. This may be part of the adaptive response mechanism towards drought stress-mediated N deficiency.

### 3.2. AtPYD1 May Play a Role in Mediating Nitrogen Assimilation through Regulation of Glutamate Synthase Encoded Genes under Drought Stress

Abiotic stress such as drought deprives plants not only from water availability but also from acquiring essential nutrients, including nitrogen (N) and N-containing compounds, which are necessary for efficient plant growth and development [18]. It has been established that glutamate synthase (protein) plays a key role as the substrate for glutamine synthase in the initial steps of N assimilation by plants [54]. In this study, we observed that the two genes encoding glutamate synthase in Arabidopsis were differentially regulated by drought stress. *AtGLU1* was downregulated, while *AtGLU2* exhibited an opposite pattern in Col-0 and *atbzip62*, and *atpyd1-2* (Figure 3E,F). We could then say that *AtGLU2* would prevail over *AtGLU1* during NO_3_ assimilation events when plants experience drought stress. In addition, we observed that *AtGLU1* and *AtGLU2* were similarly regulated in Col-0 and *atbzip62*, which may imply that *AtbZIP62* TF would not be required in the regulation of the expression of glutamate synthase-encoding genes. The KEGG analysis revealed that carbamoyl phosphate serves a precursor for de novo pyrimidine biosynthesis and the N metabolic pathways [55]. In the same way, the panel G of Figure 1 proposes that a protein-protein interaction would exist between PYD1 and CARB. Therefore, the significant increase in the transcript accumulation of *AtGLU1* and the decrease of *AtGLU2* recorded in *atpyd1-2* would suggest that *AtPYD1* might be involved in the regulation of glutamate synthase-mediated NO_3_ assimilation in plants under drought stress.

## 4. Materials and Methods

### 4.1. Plant Materials, Growth Conditions, and Drought Stress Induction

Seeds of Arabidopsis Col-0 (wild type, WT), mutant lines *atbzip62* (AT1G19490: SALK_053908C) and *atpyd1-2* (AT3G17810: SALK_083897C) were acquired from the Arabidopsis Biological Resource Center (ABRC) (https://abrc.osu.edu/, accessed on 20 March 2019). All Arabidopsis mutant lines were from a Col-0 genetic background.

Plants were challenged with drought stress at the rosette stage by employing the water withholding mehtod as described earlier by Harb and Pereira [56], with slight modifications. Briefly, soil moisture content (MC) was evaluated by monitoring the weight of each pot for each Arabidopsis genotype (in triplicate) regularly in order to track water loss. The percentage (about 30% soil MC) was calculated as a percentage of the actual weight loss to the initial weight of the saturated soil considered as 100% MC. Leaf samples were collected nine (9) days after water withholding (when symptoms of loss of turgidity and wilting of leaves were apparent). The target genes used in the study for qPCR (real-time quantitative polymerase chain reaction) analysis were selected based on their affinity for NO_3_ transport in plants. These included *AtNPF6.2/NRT1.4* (low-affinity NO_3_ transporter [6]), *AtNPF6.3*/*NRT1.1* (dual-affinity NO_3_ transporter [7,8]), *AtNRT2.1* and *AtNRT2.2* (considered as major components of the high-affinity uptake system) [9,10]. In addition, two genes encoding glutamate synthase were included to investigate their role as key substrates for glutamine synthase during N assimilation.

### 4.2. In silico Transcritpion Factor Binding Sites Analysis, Protein–Protein Interaction, and KEGG Pathways Analysis

To understand the relationship between the de novo pyrimidine biosynthesis pathway and the N metabolism in plants, the Kyoto Encyclopedia of Genes and Genomes (KEGG) was employed (https://www.genome.jp/kegg/, accessed on 11 September 2021). In addition, we were interested to see how the target NO_3_ transporters and glutamate synthase would interact with each other. Therefore, we conducted an in silico functional protein association network analysis using the STRING database (https://string-db.org/, accessed on 9 September 2021). Furthermore, the transcription regulation prediction was performed using the PlantRegMap feature within the Plant Transcription Regulatory Map (http://plantregmap.gao-lab.org/binding_site_prediction_result.php, accessed on 9 September 2021). The DNA sequences (coding sequence, CDS, FASTA format: “>ATxGxxxxx” proceeds the sequence) of target genes used for the prediction of binding sites specific to bZIP TFs were downloaded from the Arabidopsis Information Resource database (TAIR, https://www.arabidopsis.org/index.jsp, accessed on 9 September 2021).

### 4.3. Total RNA Isolation, cDNA Synthesis, and qPCR Analysis

Total RNA was isolated from leaf tissue samples using the TRI-Solution^TM^ Reagent (Cat. No: TS200-001, Virginia Tech Bio-Technology, Lot: 337871401001, Blacksburg, VA, USA) as described by the manufacturer’s protocol. Thereafter, the complementary DNA (cDNA) was synthesized as described earlier by Mun, et al. [57]. Briefly, 1 µg of RNA was used to synthesize cDNA using BioFACT^TM^ RT-Kit (BioFACT^TM^, Daejeon, Korea) according to the manufacturer’s protocol. The cDNA was then used as a template in qPCR to study the transcript accumulation of selected genes (Table 3).

The expression of genes was monitored by qPCR under the following conditions: we prepared a reaction mixture composed of SYBR green 2X Master Mix (BioFACT, Daejeon, Korea) along with 100 ng of template DNA and 10 nM of each forward and reverse primers in a final reaction of 20 µL volume. A no-template control (NTC) was used as a control. A 2-step reaction including polymerase activation at 95 °C for 15 min, following denaturation at 95 °C for 5 s, annealing and extension at 65 °C for 30 s was performed in a real-time PCR machine (Eco™ Illumina). The number of total reaction cycles was 40. Prior to assessing the changes in the transcript accumulation of the target genes, their relative expression values were normalized to that of the housekeeping gene *AtActin2*.

### 4.4. Statistical Analysis

All the experiments were conducted using a CRD design. The data were collected in triplicate and analyzed statistically with GraphPad Prism software (Version 7.00, ©1992–2016 GraphPad Software, Inc., San Diego, CA, USA). The analysis of variance (ANOVA) for Completely Randomized Design was performed, and the Turkey’s multiple comparisons test was employed at a significance level of 0.05. To assess the statistical significance level of the observed changes in the expression of target genes between Arabidopsis genotypes, we compared the normalized relative expression values in the mutant lines *atbzip62* or *atpyd1-2* with those recorded in Col-0 upon drought stress.

## 5. Conclusions

A growing interest in plant biosciences-related studies in investigating nitrogen (N) metabolism in response to abiotic stress has marked the last two decades. The use of N in Agriculture is indispensable for efficient food production and quality. However, under abiotic stress conditions, N availability to plants is compromised, which may lead to crop failure.

This study investigated the role of *AtbZIP62* encoding transcription factor and *AtPYD1* in the regulation of genes encoding low-, dual- and high-affinity NO_3_ transporters as well as glutamate synthase in response to drought stress. The results revealed that the transcript accumulation of *AtNPF6.2/NRT1.4*, *AtNPF6.3/NRT1.1*, and *AtNRT2.2* decreased significantly in *atbzip62* and *atpyd1-2* compared to Col-0 (WT). Meanwhile, *AtNRT2.1* showed a similar expression pattern in Col-0 and *atbzip62*, while having an opposite pattern in *atpyd1-*2. In the same way, glutamate synthase-encoding genes *AtGLU1* and *AtGU2* were differentially regulated by drought stress, with *AtGLU2* being upregulated. Of all NO_3_ transporter-related genes, *AtNRT2.2*, which exhibited the highest transcript accumulation level, while being differentially regulated between Col-0 and *atbzip62*, may prevail over other test genes under drought stress, under the regulatory influence of *AtbZIP62* TF. Furthermore, *AtbZIP62* may require *AtbZIP18* and/or *AtbZIP69* to regulate the expression of the analyzed NO_3_ transporters under drought stress conditions. Future studies using NO_3_ transporter-specific mutant lines, coupled with advanced physiological, biochemical and molecular analyses would help to elucidate the mechanism underlying NO_3_ transport during abiotic stress to optimize the nitrogen use efficiency in plants.

## Figures and Tables

**Figure 1 plants-10-02149-f001:**
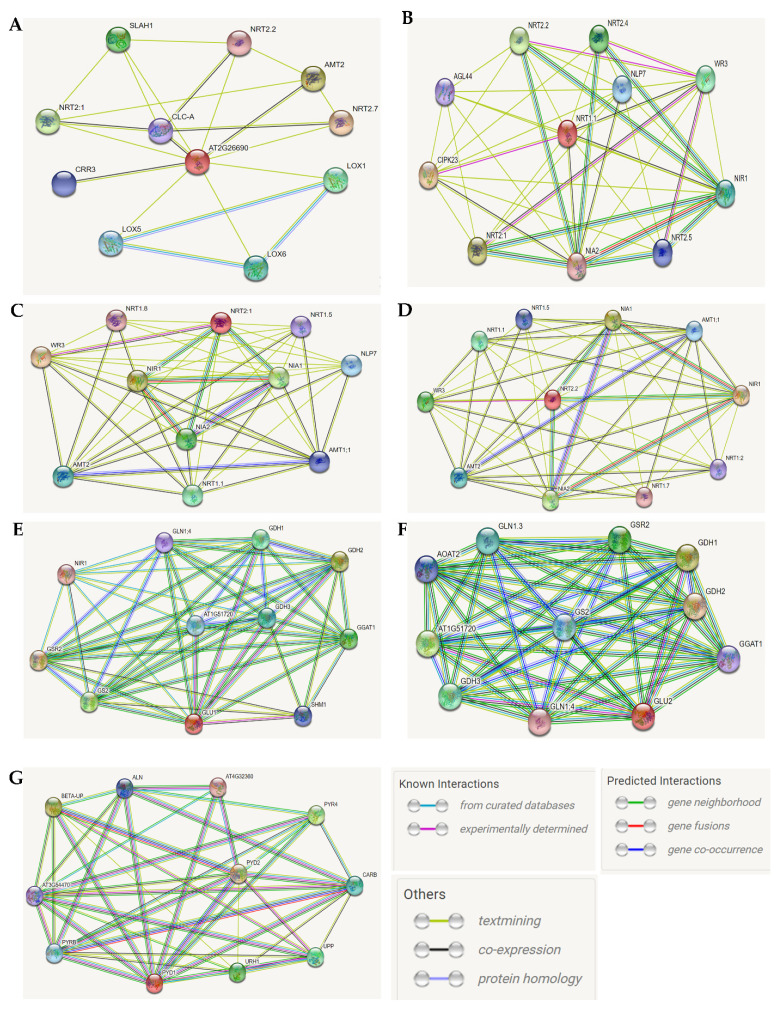
Prediction of the functional protein–protein interactome of selected nitrate transporters and assimilation-related genes. Predicted protein–protein interaction network involving the Arabidopsis (**A**) NPF6.2, (**B**) NFP6.3/NRT1.1, (**C**) NRT2.1, (**D**) NRT2.2, (**E**) GLU1, (**F**) GLU2, and (**G**) PYD1. The prediction was done using the protein sequence of each target gene obtained from the Arabidopsis Information Resource (TAIR, www.arabidopsis.org, accessed on 9 September 2021) and uploaded to STRING protein database version 11.5 (https://string-db.org/, accessed on 9 September 2021). Red nodes represent the gene of interest, while other nodes show the interactome of a different nature indicated by connecting lines. The nature of the interaction is indicated by specific line colors and the source of information (from curated databases, experimentally determined, predicted interactions, and others) as explained in the figure legend next to the panel (**G**).

**Figure 2 plants-10-02149-f002:**
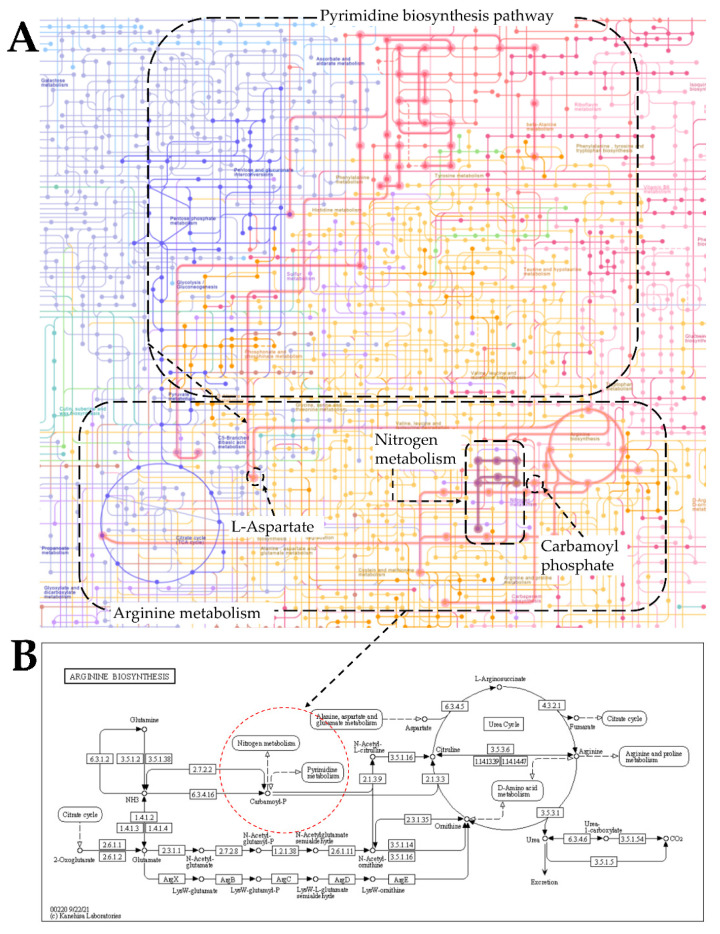
Interplay between pyrimidine, arginine nitrogen metabolism. (**A**) Shows Carbamoyl phosphate as the major substrate linking the pyrimidine pathway, arginine pathway and the nitrogen metabolism. (**B**) Displays the Arginine biosynthesis pathway highlighting the crosstalk between nitrogen metabolism and pyrimidine biosynthesis pathway (see dotted read circle). We generated this metabolic pathway using the Kyoto Encyclopedia of Genes and Genomes (KEGG) database (https://www.genome.jp/pathway/map01100+M00052, accessed on 11 September 2021).

**Figure 3 plants-10-02149-f003:**
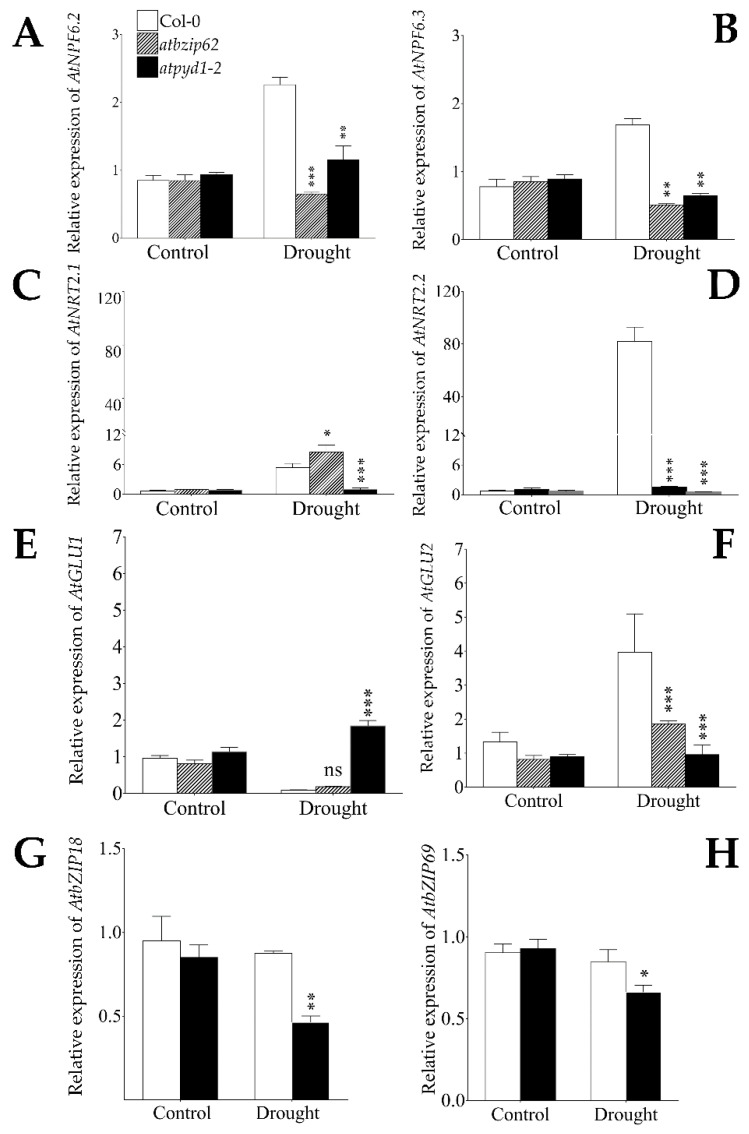
Transcript accumulation of low-, dual-, and high-affinity nitrate transporters and glutamate synthase-encoding genes in response to drought stress monitored in Arabidopsis Col-0, *atbzip62* and *atpyd1-2* mutant lines lacking *AtbZIP62* and *AtPYD1*, respectively, under drought stress. (**A**) Expression levels of *AtNFP6.2*, (**B**) *AtNPF6.3*, (**C**) *AtNRT2.1*, (**D**) *AtNRT2.2*, (**E**) *AtGLU1*, (**F**) *AtGLU2*, (**G**) *AtbZIP18*, and (**H**) *AtbZIP69* upon drought stress induction. Bars are mean values of triplicate ± SD from non-stressed and drought-stressed plants. White bars are Col-0 WT, bars with hatch lines represent the *atbzip62*, and black bars indicate the *atpyd1-2*. The relative expression values of each target gene were normalized to that of the housekeeping gene *Actin2*. For statistical significance, the normalized transcript accumulation level of target genes in the mutant lines was compared with that recorded in Col-0 wild type. The statistical significance is indicated on the top of each bar. * *p* < 0.05, ** *p* < 0.01, *** *p* < 0.001, ns non-significant.

**Figure 4 plants-10-02149-f004:**
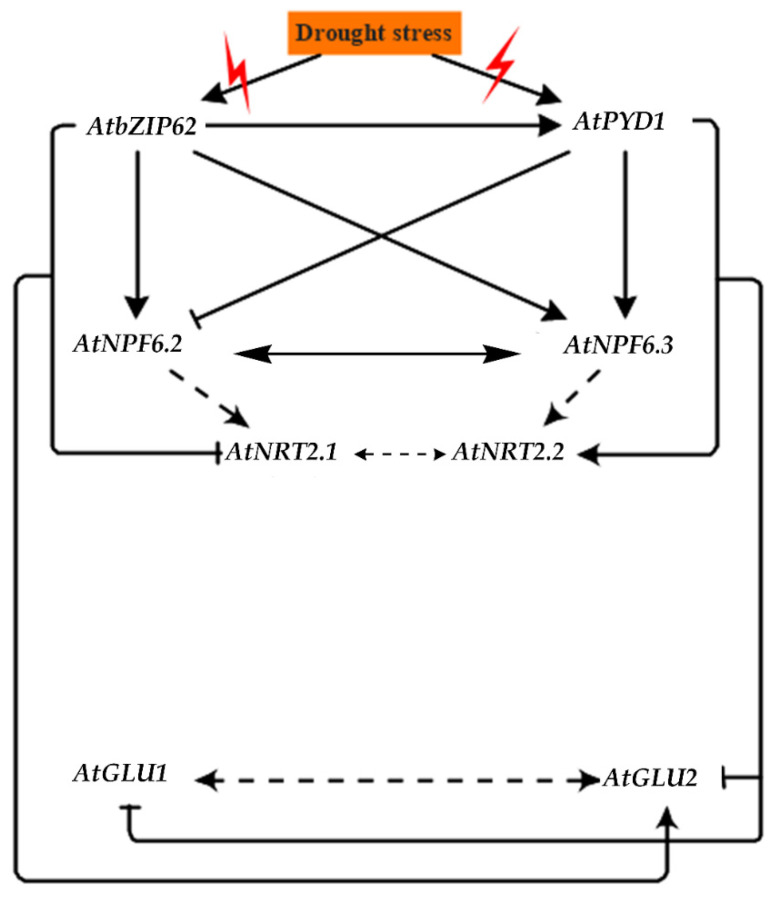
Signaling model of the transcriptional regulation of nitrate transporters as well as glutamate synthase-encoding genes in response to drought stress. When plants experience adverse environmental conditions, they activate signaling cascades and the appropriate defense system, which includes the induction of a wide range of drought-responsive genes, of which the transcript accumulation events are mediated by transcription factors, and their interplay determines the degree of the defense. The basic leucine zipper (bZIP) transcription factor *AtbZIP62* TF is here suggested to interact with other proteins or DNA to regulate the expression of *AtNPF6.2* (low-affinity NO_3_ transporter), *AtNPF6.3*/*NRT1.1* (dual-affinity NO_3_ transporter), *AtNRT2.1* and *AtNRT2.2* (both having a high affinity for NO_3_ transport), as well as *AtGLU1* and *AtGLU2* (encoding glutamate synthase) under drought stress conditions in Arabidopsis. Arrows with continuous lines indicate a positive regulation (of gene expression); whereas, continuous lines with a perpendicular bar (blunt end) show a negative regulation of gene expression by our studies (current and previous). Dotted lines with an arrow at their tips suggest a possible functional interaction as predicted through in silico transcription factor binding sites analysis and protein–protein interaction, coupled with the literature. Dotted lines with arrows indicate a proposed interaction between NO_3_ transport and assimilation in the literature.

**Table 1 plants-10-02149-t001:** Identified transcription factor binding sites in the promoter of target genes.

Gene Locus ID	TF Name	Target Genes	Position	Strand	*p*-Value	q-Value	Matched Sequence
*AtbZIP62* TF							
AT1G06070	AtbZIP69	AT1G19490	1528–1538	−	3.77 × 10^−5^	0.154	AACAACTGGCC
AT2G40620	AtbZIP18	AT1G19490	1912–1922	+	1.31 × 10^−5^	0.0546	ATGAGCTGGCA
*AtPYD1*							
AT1G06070	AtbZIP69	AT3G17810	427–437	+	2.8 × 10^−5^	0.115	TGCAGCTGGAG
AT1G06070	AtbZIP69	AT3G17810	427–437	−	6.13 × 10^−5^	0.126	TGCAGCTGTTG
*AtNPF6.2/NRT1.4*							
AT1G06070	AtbZIP69	AT2G26690	2934–2944	−	1.04 × 10^−5^	0.079	ACCAGCTGGGA
AT1G06070	AtbZIP69	AT2G26690	2935–2945	+	4.24 × 10^−5^	0.109	CCCAGCTGGTT
AT1G06070	AtbZIP69	AT2G26690	1096–1106	−	4.34 × 10^−5^	0.109	GACAACTGGTA
AT2G40620	AtbZIP18	AT2G26690	2983–2993	+	7.12 × 10^−5^	0.34	TCTAGCTGTCT
AT2G40620	AtbZIP18	AT2G26690	3469–3479	+	8.91 × 10^−5^	0.34	GATGGCTGGCT
*AtNPF6.3*/*NRT1.1*							
*	*	AT1G12110	*	*	*	*	*
*	*	AT1G12110	*	*	*	*	*
*AtNRT2.1*							
*	*	AT1G08090	*	*	*	*	*
*	*	AT1G08090	*	*	*	*	*
*AtNRT2.2*							
AT1G06070	AtbZIP69	AT1G08100	1761–1771	−	9.28 × 10^−5^	0.388	GACAACTGTAT
*AtGLU1*							
AT2G40620	AtbZIP18	AT5G04140	459–469	+	8.32 × 10^−5^	0.348	TAGGGCTGGCT
*AtGLU2*							
*	*	AT2G41220	*	*	*	*	*
*	*	AT2G41220	*	*	*	*	*

(*) the asterisk indicates that the binding sites for the *AtbZIP18* and/or *AtbZIP69* associated with *AtbZIP62* TF were not detected.

**Table 2 plants-10-02149-t002:** Predicted protein–protein interactions involving *AtNRT2.2* (topmost upregulated by drought stress).

Target Proteins	Locus	Description	Reference
NIR1	AT2G15620	Ferredoxin—nitrite reductase, chloroplastic; Involved in the second step of nitrate assimilation.	[28]
NIA1	AT1G12110	Nitrate reductase [NADH] 1; Encodes the cytosolic minor isoform of nitrate reductase (NR). Involved in the first step of nitrate assimilation, it contributes about 15% of the nitrate reductase activity in shoots.	[29]
NIA2	AT1G37130	Nitrate reductase [NADH] 2; Identified as a mutant resistant to chlorate. Encodes nitrate reductase structural gene. Involved in nitrate assimilation. Has nitrate reductase activity.	[29]
WR3	AT1G08100	High-affinity nitrate transporter 3.1; Acts as a dual component transporter with NTR2.1. Required for high-affinity nitrate transport. May be involved in targeting NRT2 proteins to the plasma membrane.	[30]
NRT1.1	AT1G12110	Protein NRT1/PTR family 6.3; Dual affinity nitrate transporter. Involved in proton- dependent nitrate uptake and in the regulation of the nitrate transporter NRT2.1. Acts also as a nitrate sensor that trigger a specific signaling pathway stimulating lateral root growth and seed germination. The uptake activity is not required for sensor function.	[31]
NRT1.5	AT1G32450	Protein NRT1/PTR FAMILY 7.3; Transmembrane nitrate transporter. Involved in xylem transport of nitrate from root to shoot. Induced in response to nitrate. Not involved in nitrate uptake. Belongs to the PTR2/POT transporter (TC 2.A.17) family	[33]
NRT1.2	AT1G69850	Protein NRT1/PTR family 4.6; Low-affinity proton-dependent nitrate transporter. Involved in constitutive nitrate uptake. Involved in (+)-abscisic acid (ABA) transport. Mediates cellular ABA uptake. Belongs to the PTR2/POT transporter (TC 2.A.17) family	[32]
NRT1.7	AT1G69870	Protein NRT1/PTR family 2.13; Low-affinity proton-dependent nitrate transporter. Involved in phloem loading and nitrate remobilization from the older leaves to other tissues; Belongs to the PTR2/POT transporter (TC 2.A.17) family	[34]

**Table 3 plants-10-02149-t003:** Arabidopsis mutant lines and set of qPCR primers for the expression of target genes used in the study.

Gene Name/ Genotype	Locus/SALK	5′-Forwad Primer-3′	5′-Reverse Primer-3′	GC Content (%) F/R	Amplicon Size (bp)
Genotyping primers (Left border and right border)					
*atbzip62*	SALK_053908C	TGGCACTTTTAACTTTGTGCC	TACGTTTCCATCGAGTGAACC	-	*atbzip62* mutant
*atpyd1-2*	SALK_083897C	TTGGGTGGCAGAACATAGAAC	ATGAATTCAGCGGCATCATAG	-	*atpyd1-2* mutant
Nitrate transporters and assimilation genes in Arabidopsis					
*AtNPF6.2*	AT2G26690	TGGAGAGCAAAGGGAGTTGG	AATGAGAGCGGCAGTGATCC	55.0/55.0	102
*AtNPF6.3*	AT1G12110	ATGAAAGGGATGAGCACGGG	CATGGATGAGCTTTCCCGGT	55.0/55.0	110
*AtNRT2.1*	AT1G08090	GGCTACGCATCTGACTTTGC	AACGGCAGTTACAAGGGTGT	55.0/50.0	132
*AtNRT2.2*	AT1G08100	CTCCGTCTCGGGGAGTATCT	TCATGGAGAACACCGTTGGG	60.0/55.0	119
*AtGLU1*	AT5G04140	CTTCTGCATGGGCGACGATA	CCTAAGGGGGTCAATGGCAG	55.0/60.0	118
*AtGLU2*	AT2G41220	GCAGCATTTAGCCAACCGTC	AGGCTCAACCTTCCCAACAG	55.0/55.0	94
*AtActin2*	AT3G18780	CGCTGACCGTATGAGCAAAG	GGAACCACCGATCCAGACAC	55.0/60.0	106

## Data Availability

Not Applicable.

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
