# Peer review of "Regulation of Nitrate (NO3) Transporters and Glutamate Synthase-Encoding Genes under Drought Stress in Arabidopsis: The Regulatory Role of AtbZIP62 Transcription Factor"

_plants, 2021, doi:10.3390/plants10102149_

Round 1
Reviewer 1 Report
The authors of the manuscript report expression data measured by qPCR in Arabidopsis plants for a number of nitrate transport and nitrogen metabolism genes as effected by the presence/absence of Atbzip62 and Atpyd1-2 in a Col-0 background.
Based on the results they concluded that AtNRT2.2, which exhibited the highest transcript accumulation level under drought stress, while being differentially regulated between Col-0 and atbzip62, may prevail over other test genes under drought stress, under the regulatory influence of AtbZIP62 TF.
The manuscript is well written and although the analysis is a relatively simple study of gene expression of a set of related genes, the results are novel and carry implications for better understanding of nitrogen availability signaling and metabolism under drought stress.
As far as the experimental design and methods go, I have two concerns.
1) One is the ambiguity in information of how expression values were used for comparison. Because the drought treatment is accompanied by a loss of tissue FW, some of the observed differences could be caused by water loss if comparisons are not done in a specific way. The authors say:
"Expression data in the mutant lines were compared with that in Col-0." and "data were normalized with relative expression of Arabidopsis Actin2."
I suggest the authors state explicitly how many and which actin measurements were used for normalization and which specific data were compared (presumably after normalization).
2) The other is the fact that some of the primers (but not primer pairs) are present in alternative transcripts of the measured genes and that some of the genes have such trasncripts. I would like the authors to critically assess the possibility that their assays may or may not recognize certain transcript forms and how this could affect the results, if at all.
MINOR
3) Apart from this I noticed that Figure 1 and 2 are presented in the wrong order in my version of the manuscript. Please, make sure that Figures are numbered in the order they are first referenced in the text and if possible, are displayed in the same order in the manuscript.
4) A few suggestions for changed wording:
"Our findings revealed that the expression of AtNPF6.2/NRT1.4, AtNPF6.3/NRT1.1 and AtNRT2.2 was similarly regulated in atzbip62 and atpyd1-2 but differentially regulated between the mutant lines Col-0"
Maybe there is an "and" missing before "Col-0"?
"Therefore(??), this study aimed to investigate the role of AtbZIP62 TF in the regulation of the expression of four (4) NO 3 transporters and two glutamate synthase encoding genes involved in N assimilation in Arabidopsis in response to drought stress."
This sentence is at a beginning of a paragraph following a paragraph on leucine zipper proteins and transcription factors. Using the word "therefore" feels inappropriate here. I do not think the presented characteristics of TFs were the reason why the authors decided to investigate AtbZIP62. Please, be more specific why the study was initiated or otherwise change the semantic linkage between the two paragraphs.
Otherwise, I find the manuscript well written, original, even if targeted toward a small group of scientists, because of the simple research question and very narrow focus. As such I recommend it for publication after minor fixes.
Author Response
Regulation of Nitrate (NO3) Transporters and Glutamate Synthase Encoding Genes under Drought Stress in Arabidopsis: The Regulatory Role of AtbZIP62 Transcription Factor
Manuscript ID: plants-1405227
Point by point response to the comments of reviewers
We are thankful to the editorial office and anonymous reviewers for their time given to evaluate manuscript. We appreciate their comments, and are happy to share that most of the comments are addressed and have substantially improved the quality of the manuscript. We would like to specify that all changes in the manuscript were highlighted yellow. We hope that the manuscript in the current form will be suitable for publication in the journal.
|
Reviewer 1 |
|
|
The authors of the manuscript report expression data measured by qPCR in Arabidopsis plants for a number of nitrate transport and nitrogen metabolism genes as effected by the presence/absence of Atbzip62 and Atpyd1-2 in a Col-0 background. Based on the results they concluded that AtNRT2.2, which exhibited the highest transcript accumulation level under drought stress, while being differentially regulated between Col-0 and atbzip62, may prevail over other test genes under drought stress, under the regulatory influence of AtbZIP62 TF. The manuscript is well written and although the analysis is a relatively simple study of gene expression of a set of related genes, the results are novel and carry implications for better understanding of nitrogen availability signaling and metabolism under drought stress. |
We are thankful to the reviewer for his valuable comments and observations that helped us substantially improve the manuscript. We have tried to address the question raised by the reviewer to the best our understanding. |
|
As far as the experimental design and methods go, I have two concerns. |
|
|
1) One is the ambiguity in information of how expression values were used for comparison. Because the drought treatment is accompanied by a loss of tissue FW, some of the observed differences could be caused by water loss if comparisons are not done in a specific way. The authors say: "Expression data in the mutant lines were compared with that in Col-0." and "data were normalized with relative expression of Arabidopsis Actin2."
I suggest the authors state explicitly how many and which actin measurements were used for normalization and which specific data were compared (presumably after normalization). |
We appreciate the concern raised by the reviewer. We would like to apologize for causing this ambiguity. Here, we intended to say that the relative expression of each target genes was normalized to that of the housekeeping gene AtAtin2. The normalization of qPCR data is a common practice, which is recommended as part of the data quality control. We have modified the statement as it appears in lines 407–408, under the subsection 4.3, paragraph 2. In addition, we have provided a modified statement to clarify at the level of comparison of expressions between drought-treated Col-0 and the mutant lines for each analyzed gene as follows: Lines 415–418: “To assess the statistical significance level of the observed changes in the expression of target genes between Arabidopsis genotypes, we compared the normalized relative expression values in the mutant lines atbzip62 or atpyd1-2 with that recorded in Col-0 upon drought stress. |
|
2) The other is the fact that some of the primers (but not primer pairs) are present in alternative transcripts of the measured genes and that some of the genes have such transcripts. I would like the authors to critically assess the possibility that their assays may or may not recognize certain transcript forms and how this could affect the results, if at all. |
We sincerely appreciate the pertinence of the concern raised by the reviewer. We would like to argue that prior to performing qPCR analysis; primers were designed to avoid redundancy if any or self-complementation of the primer. For the forward and reverse primers are selected based on their length, GC content, self-complementation value, and there were BLAST in NCBI to confirm that in Arabidopsis thaliana, no transcript from other genes than that analyzed could be amplified, which would be seen as redundancy. In our case, we did not find any redundancy in the primers for all the genes with other genes. In our case, genes selected have only one transcript type. If we refer to AT2G6690, we can see that when searching this gene in TAIR (www.arabidopsis.org), two loci are displayed; one is AT2G6690 and the other is AT2G26692.1 which is indicated as an antisense noncoding RNA. Likewise, AT1G08100 search in the same database shows other loci named
We confirm that AT5G04140 (AtGLU1) showing to have a variant named In both cases, we have noted that The reverse primer (reverse complement) matches with a region in the CDS of the variant. We would like to indicate that it is common to see genes with more than one alternative or splicing variants. Here, the variants of AtGLU1 or AtGLU2 are representing their corresponding genes. Therefore, it is not surprising that the reverse primers matches a sequence in the coding sequences (CDS) of these genes. In this case, we could not expect to have redundancy in the expression of these genes under the conditions of this study. |
|
MINOR |
|
|
3) Apart from this I noticed that Figure 1 and 2 are presented in the wrong order in my version of the manuscript. Please, make sure that Figures are numbered in the order they are first referenced in the text and if possible, are displayed in the same order in the manuscript. |
We would like to apologize for the inconvenience. We have corrected the order and the numbering of all figures in the manuscript, and their respective citations have been corrected as well. |
|
4) A few suggestions for changed wording: "Our findings revealed that the expression of AtNPF6.2/NRT1.4, AtNPF6.3/NRT1.1 and AtNRT2.2 was similarly regulated in atzbip62 and atpyd1-2 but differentially regulated between the mutant lines Col-0" Maybe there is an "and" missing before "Col-0"? |
We would like to apologize for the missing conjunction. We have added “and” in the abstract at the designated place as suggested (page 1, line 31). |
|
"Therefore(??), this study aimed to investigate the role of AtbZIP62 TF in the regulation of the expression of four (4) NO 3 transporters and two glutamate synthase encoding genes involved in N assimilation in Arabidopsis in response to drought stress."
This sentence is at a beginning of a paragraph following a paragraph on leucine zipper proteins and transcription factors. Using the word "therefore" feels inappropriate here. I do not think the presented characteristics of TFs were the reason why the authors decided to investigate AtbZIP62. Please, be more specific why the study was initiated or otherwise change the semantic linkage between the two paragraphs. |
We would like to thank the reviewer for his valuable comment. While trying to take into consideration the suggestion, we have clarified the reasons for investigating NO3 transporters and assimilation in atbzip62 and atpyd1-2 mutant lines in lines 100-117, page 3.
|
|
Otherwise, I find the manuscript well written, original, even if targeted toward a small group of scientists, because of the simple research question and very narrow focus. As such I recommend it for publication after minor fixes. |
We would like to thank the reviewer for his valuable time committed to evaluate this brief report. All the comments given to us have helped us improve the manuscript. |
AT1G08100 (AtNRT2.2 used in our study)
|
Gene |
Locus ID |
Forward primer (5’->3’) |
Reverse primer (5’->3’) |
|
|
AtNRT2.2 |
AT1G08100 |
CTCCGTCTCGGGGAGTATCT |
TCATGGAGAACACCGTTGGG |
Nitrate transporter 2.2. |
Reverse complement of the reverse primer : 5’- CCCAACGGTGTTCTCCATGA-3’
CDS of AT1G08100 (AtNRT2.2 used in our study)
ATGGGTTCTACTGATGAGCCCGGAAGTTCCATGCATGGAGTTACCGGTAGAGAACAGAGCTATGCTTTCTCGGTAGATGGTAGTGAGCCGACCAACACAAAGAAAAAGTACAATCTGCCGGTGGACGCGGAGGATAAGGCAACGGTTTTCAAGCTCTTCTCCTTCGCCAAACCTCACATGAGAACGTTCCACCTCTCGTGGATCTCTTTCTCCACATGTTTTGTTTCGACGTTCGCAGCTGCACCACTTATCCCGATCATCAGGGAGAATCTTAACCTCACCAAACATGACATTGGAAACGCTGGAGTTGCCTCCGTCTCGGGGAGTATCTTCTCTAGGCTCGTGATGGGAGCCGTGTGTGATCTTTTGGGTCCTCGTTACGGTTGTGCCTTCCTTGTGATGTTGTCTGCCCCAACGGTGTTCTCCATGAGCTTCGTGAGTGACGCAGCAGGCTTCATAACGGTGAGGTTCATGATTGGTTTTTGCCTGGCGACGTTTGTGTCTTGTCAATACTGGATGAGCACTATGTTCAACAGTCAGATCATCGGTCTGGTGAACGGGACAGCAGCCGGATGGGGAAACATGGGTGGCGGCATAACGCAGTTGCTCATGCCCATTGTGTATGAAATCATTAGGCGCTGCGGATCAACAGCGTTCACGGCCTGGAGGATCGCCTTCTTTGTCCCCGGTTGGTTGCACATCATCATGGGAATCTTGGTGCTCACGCTAGGTCAAGATCTGCCAGGTGGAAACAGAGCTGCCATGGAGAAAGCGGGAGAAGTTGCCAAAGACAAATTCGGAAAGATTCTATGGTACGCCGTTACAAATTACAGGACTTGGATTTTCGTTCTTCTGTATGGATATTCCATGGGAGTTGAGTTAAGCACAGACAATGTTATCGCCGAGTACTTCTTTGATAGGTTTCACTTGAAGCTTCACACAGCGGGGATTATAGCAGCATGTTTCGGAATGGCCAATTTCTTTGCTCGTCCAGCAGGAGGCTGGGCATCTGACATTGCAGCCAAGCGCTTCGGAATGCGAGGGAGGTTGTGGACTTTGTGGATCATTCAGACGTCCGGTGGTCTCTTTTGTGTGTGGCTCGGACGTGCCAACACCCTCGTCACTGCCGTTGTATCTATGGTCCTCTTCTCTTTAGGAGCACAAGCCGCTTGCGGAGCCACCTTTGCTATCGTGCCCTTTGTCTCCCGGCGAGCTCTAGGCATTATCTCGGGTTTAACCGGGGCTGGAGGGAACTTTGGGTCAGGACTCACACAGCTCGTCTTTTTCTCGACTTCGCGCTTCACAACTGAAGAAGGGCTAACGTGGATGGGAGTGATGATAGTTGCTTGCACGTTGCCTGTTACCTTAATCCACTTTCCTCAGTGGGGAAGCATGTTCTTCCCTCCTTCCAACGATTCGGTCGACGCTACGGAGCACTATTATGTTGGCGAATATAGTAAGGAGGAGCAGCAGATTGGCATGCATTTAAAAAGCAAACTGTTTGCTGATGGAGCCAAGACCGAGGGAGGCAGCAGCGTCCACAAAGGGAACGCAACCAACAATGCTTGA
--------
CDS of other locus FGENESH2_KG.1__820__AT1G08100.1.1
ATGGGTGATTCTAATGGTGAACCCGGAAGCTCCATGCATGGAGTTACCGGTAGAGAACAGAGCTATGCTTTCTCGGTAGATACACCAACTGTGCCGACCAACACAAAGAAAAAGTACAACCTGCCGGTGGATGCAGAGGATAAGGCAACGGTTTTCAAGCTCTTCTCCTTCGCAAAACCTCATATGAGAACGTTCCACCTCTCGTGGATCTCTTTCTCAACATGTTTCGTCTCGACATTCGCTGCTGCACCACTTATCCCGATCATCAGGGAGAATCTCAACCTCACCAAACAAGACATTGGAAACGCAGGAGTTGCCTCTGTCTCTGGGAGTATTTTTTCTAGGCTGGTGATGGGAGCCGTGTGTGATCTTTTGGGTCCTCGTTACGGCTGTGCCTTCCTTGTGATGCTGTCTGCGCCAACGGTGTTCTCCATGAGTTTCGTGAGTGGAGCCGCAGGCTTCATAACGGTGAGGTTCATGATTGGTTTCTGCCTGGCGACGTTTGTGTCTTGTCAATACTGGATGAGCACTATGTTCACCAGTCAGATCATCGGTCTGGTAAACGGGACAGCAGCCGGATGGGGAAACATGGGTGGCGGCATAACGCAGTTGCTCATGCCTATTGTGTATGAAATCATTAGGCGTTGCGGTTCAACAGCGTTCACGGCCTGGAGGATCGCCTTCTTTGTCCCCGGTTGGTTGCACATCATTATGGGAATCTTGGTGCTCACGCTAGGTCAAGATCTGCCAGGTGGAAACCGAGCTGCCATGGAGAAATCGGGAGAAGTTGCCAAAGACAAATTCGGAAAGATTCTGTGGTATGCCGTTACAAATTACAGGACTTGGATTTTCGTTCTTCTGTATGGATATTCCATGGGAGTTGAGTTAAGCACTGACAATGTTATCGCCGAGTACTTCTTTGACAGGTTTCACTTGAAACTTCACACAGCGGGGATCATAGCAGCGTGTTTCGGGATGGCCAATTTCTTTGCTCGTCCAGCAGGAGGCTGGGCATCTGACATTGCAGCCAGGCGCTTCGGGATGAGAGGGAGGTTGTGGACGTTGTGGATCATTCAGACGGCCGGTGGTCTCTTTTGTGTGTGGCTCGGTCGTGCCAACACGCTCGTCACTGCCGTTGTATCTATGGTCCTCTTCTCTTTAGGAGCACAAGCCGCTTGTGGAGCCACCTTTGCAATCGTGCCCTTTGTCTCCCGGAGAGCTCTAGGCATTATCTCTGGTTTAACCGGAGCTGGAGGCAATTTTGGGTCTGGACTCACACAACTCGTCTTCTTCTCGACTTCACGCTTCACAACTGAAGAAGGGCTAACGTGGATGGGAGTGATGATAGTTGCTTGCACGTTGCCTGTGACCTTAATCCACTTTCCTCAATGGGGAAGCATGTTCTTGCCTCCCTCCAACGATTCGATCGATGCCACGGAGCATTATTATGTTTCCGAGTATAATAAGCAGGAGCAGCAGGATGGCATGCATTTAAAAAGCATCAAGTTTGCTGAGGGAGCCAGGACCGAGGGCGGCCCCAGCGTCCACAATGGGAACATTTCCAACAATGCTTGA
CLUSTAL O(1.2.1) multiple sequence alignment of AT1G08100 and At1G08100.1.1
AT1G08100 ---ATGGGTTCTACTGATGAGCCCGGAAGTTCCATGCATGGAGTTACCGGTAGAGAACAG
AT1G08100.1.1 ATGGGTGATTCTAATGGTGAACCCGGAAGCTCCATGCATGGAGTTACCGGTAGAGAACAG
* ***** ** *** ******** ******************************
AT1G08100 AGCTATGCTTTCTCGGTAGATGGT---AGTGAGCCGACCAACACAAAGAAAAAGTACAAT
AT1G08100.1.1 AGCTATGCTTTCTCGGTAGATACACCAACTGTGCCGACCAACACAAAGAAAAAGTACAAC
********************* * ** ***************************
AT1G08100 CTGCCGGTGGACGCGGAGGATAAGGCAACGGTTTTCAAGCTCTTCTCCTTCGCCAAACCT
AT1G08100.1.1 CTGCCGGTGGATGCAGAGGATAAGGCAACGGTTTTCAAGCTCTTCTCCTTCGCAAAACCT
*********** ** ************************************** ******
AT1G08100 CACATGAGAACGTTCCACCTCTCGTGGATCTCTTTCTCCACATGTTTTGTTTCGACGTTC
AT1G08100.1.1 CATATGAGAACGTTCCACCTCTCGTGGATCTCTTTCTCAACATGTTTCGTCTCGACATTC
** *********************************** ******** ** ***** ***
AT1G08100 GCAGCTGCACCACTTATCCCGATCATCAGGGAGAATCTTAACCTCACCAAACATGACATT
AT1G08100.1.1 GCTGCTGCACCACTTATCCCGATCATCAGGGAGAATCTCAACCTCACCAAACAAGACATT
** *********************************** ************** ******
AT1G08100 GGAAACGCTGGAGTTGCCTCCGTCTCGGGGAGTATCTTCTCTAGGCTCGTGATGGGAGCC
AT1G08100.1.1 GGAAACGCAGGAGTTGCCTCTGTCTCTGGGAGTATTTTTTCTAGGCTGGTGATGGGAGCC
******** *********** ***** ******** ** ******** ************
AT1G08100 GTGTGTGATCTTTTGGGTCCTCGTTACGGTTGTGCCTTCCTTGTGATGTTGTCTGCCCCA
AT1G08100.1.1 GTGTGTGATCTTTTGGGTCCTCGTTACGGCTGTGCCTTCCTTGTGATGCTGTCTGCGCCA
***************************** ****************** ******* ***
AT1G08100 ACGGTGTTCTCCATGAGCTTCGTGAGTGACGCAGCAGGCTTCATAACGGTGAGGTTCATG
AT1G08100.1.1 ACGGTGTTCTCCATGAGTTTCGTGAGTGGAGCCGCAGGCTTCATAACGGTGAGGTTCATG
***************** ********** ** ***************************
AT1G08100 ATTGGTTTTTGCCTGGCGACGTTTGTGTCTTGTCAATACTGGATGAGCACTATGTTCAAC
AT1G08100.1.1 ATTGGTTTCTGCCTGGCGACGTTTGTGTCTTGTCAATACTGGATGAGCACTATGTTCACC
******** ************************************************* *
AT1G08100 AGTCAGATCATCGGTCTGGTGAACGGGACAGCAGCCGGATGGGGAAACATGGGTGGCGGC
AT1G08100.1.1 AGTCAGATCATCGGTCTGGTAAACGGGACAGCAGCCGGATGGGGAAACATGGGTGGCGGC
******************** ***************************************
AT1G08100 ATAACGCAGTTGCTCATGCCCATTGTGTATGAAATCATTAGGCGCTGCGGATCAACAGCG
AT1G08100.1.1 ATAACGCAGTTGCTCATGCCTATTGTGTATGAAATCATTAGGCGTTGCGGTTCAACAGCG
******************** *********************** ***** *********
AT1G08100 TTCACGGCCTGGAGGATCGCCTTCTTTGTCCCCGGTTGGTTGCACATCATCATGGGAATC
AT1G08100.1.1 TTCACGGCCTGGAGGATCGCCTTCTTTGTCCCCGGTTGGTTGCACATCATTATGGGAATC
************************************************** *********
AT1G08100 TTGGTGCTCACGCTAGGTCAAGATCTGCCAGGTGGAAACAGAGCTGCCATGGAGAAAGCG
AT1G08100.1.1 TTGGTGCTCACGCTAGGTCAAGATCTGCCAGGTGGAAACCGAGCTGCCATGGAGAAATCG
*************************************** ***************** **
AT1G08100 GGAGAAGTTGCCAAAGACAAATTCGGAAAGATTCTATGGTACGCCGTTACAAATTACAGG
AT1G08100.1.1 GGAGAAGTTGCCAAAGACAAATTCGGAAAGATTCTGTGGTATGCCGTTACAAATTACAGG
*********************************** ***** ******************
AT1G08100 ACTTGGATTTTCGTTCTTCTGTATGGATATTCCATGGGAGTTGAGTTAAGCACAGACAAT
AT1G08100.1.1 ACTTGGATTTTCGTTCTTCTGTATGGATATTCCATGGGAGTTGAGTTAAGCACTGACAAT
***************************************************** ******
AT1G08100 GTTATCGCCGAGTACTTCTTTGATAGGTTTCACTTGAAGCTTCACACAGCGGGGATTATA
AT1G08100.1.1 GTTATCGCCGAGTACTTCTTTGACAGGTTTCACTTGAAACTTCACACAGCGGGGATCATA
*********************** ************** ***************** ***
AT1G08100 GCAGCATGTTTCGGAATGGCCAATTTCTTTGCTCGTCCAGCAGGAGGCTGGGCATCTGAC
AT1G08100.1.1 GCAGCGTGTTTCGGGATGGCCAATTTCTTTGCTCGTCCAGCAGGAGGCTGGGCATCTGAC
***** ******** *********************************************
AT1G08100 ATTGCAGCCAAGCGCTTCGGAATGCGAGGGAGGTTGTGGACTTTGTGGATCATTCAGACG
AT1G08100.1.1 ATTGCAGCCAGGCGCTTCGGGATGAGAGGGAGGTTGTGGACGTTGTGGATCATTCAGACG
********** ********* *** **************** ******************
AT1G08100 TCCGGTGGTCTCTTTTGTGTGTGGCTCGGACGTGCCAACACCCTCGTCACTGCCGTTGTA
AT1G08100.1.1 GCCGGTGGTCTCTTTTGTGTGTGGCTCGGTCGTGCCAACACGCTCGTCACTGCCGTTGTA
**************************** *********** ******************
AT1G08100 TCTATGGTCCTCTTCTCTTTAGGAGCACAAGCCGCTTGCGGAGCCACCTTTGCTATCGTG
AT1G08100.1.1 TCTATGGTCCTCTTCTCTTTAGGAGCACAAGCCGCTTGTGGAGCCACCTTTGCAATCGTG
************************************** ************** ******
AT1G08100 CCCTTTGTCTCCCGGCGAGCTCTAGGCATTATCTCGGGTTTAACCGGGGCTGGAGGGAAC
AT1G08100.1.1 CCCTTTGTCTCCCGGAGAGCTCTAGGCATTATCTCTGGTTTAACCGGAGCTGGAGGCAAT
*************** ******************* *********** ******** **
AT1G08100 TTTGGGTCAGGACTCACACAGCTCGTCTTTTTCTCGACTTCGCGCTTCACAACTGAAGAA
AT1G08100.1.1 TTTGGGTCTGGACTCACACAACTCGTCTTCTTCTCGACTTCACGCTTCACAACTGAAGAA
******** *********** ******** *********** ******************
AT1G08100 GGGCTAACGTGGATGGGAGTGATGATAGTTGCTTGCACGTTGCCTGTTACCTTAATCCAC
AT1G08100.1.1 GGGCTAACGTGGATGGGAGTGATGATAGTTGCTTGCACGTTGCCTGTGACCTTAATCCAC
*********************************************** ************
AT1G08100 TTTCCTCAGTGGGGAAGCATGTTCTTCCCTCCTTCCAACGATTCGGTCGACGCTACGGAG
AT1G08100.1.1 TTTCCTCAATGGGGAAGCATGTTCTTGCCTCCCTCCAACGATTCGATCGATGCCACGGAG
******** ***************** ***** ************ **** ** ******
AT1G08100 CACTATTATGTTGGCGAATATAGTAAGGAGGAGCAGCAGATTGGCATGCATTTAAAAAGC
AT1G08100.1.1 CATTATTATGTTTCCGAGTATAATAAGCAGGAGCAGCAGGATGGCATGCATTTAAAAAGC
** ********* *** **** **** *********** *******************
AT1G08100 AAACTGTTTGCTGATGGAGCCAAGACCGAGGGAGGCAGCAGCGTCCACAAAGGGAACGCA
AT1G08100.1.1 ATCAAGTTTGCTGAGGGAGCCAGGACCGAGGGCGGCCCCAGCGTCCACAATGGGAACATT
* ********* ******* ********* *** ************ ******
AT1G08100 ACCAACAATGCTTGA
AT1G08100.1.1 TCCAACAATGCTTGA
**************
CLUSTAL O(1.2.1) multiple sequence alignment of AtGLU1 and its splicing variant.
|
Gene |
Locus |
5'-Forwad primer -3' |
5'-Reverse primer -3' |
GC content (%) F/R |
Amplicon size (bp) |
|
AtGLU1 |
AT5G04140 |
CTTCTGCATGGGCGACGATA |
CCTAAGGGGGTCAATGGCAG |
55.0/60.0 |
118 |
AT5G04140_-_AtGLU1 ATGGCGATGCAATCTCTTTCCCCTGTTCCTAAGCTTCTCTCCACAACACCAAGCTCTGTT
AT5G04140.1_variant_of_AtGLU1 ATGGCGATGAAATCTCTTTCCCCTATTCCTAAGCTTCTCTCCACAACACCAAGCTCTGTT
********* ************** ***********************************
AT5G04140_-_AtGLU1 CTTTCTTCTGACAAGAACTTCTTCTTCGTCGATTTCGTTGGATTATACTGTAAGTCCAAG
AT5G04140.1_variant_of_AtGLU1 CTTTCTTCTGACAAGAACTTCTTCTTCGTCGATTTCGTTGGATTATACTGTAAGTCCAAG
************************************************************
AT5G04140_-_AtGLU1 AGGACCAGACGCAGACTTCGTGGAGACTCTTCCTCTTCCTCACGCTCTTCTTCTTCTCTC
AT5G04140.1_variant_of_AtGLU1 AGAACCAGACGCAGACTTCGTGGCGATTCCTCCTCAGGC---TCTTCTTCTTCTTCTCTC
** ******************** ** ** ***** * ***************
AT5G04140_-_AtGLU1 TCTCGTCTTTCCTCTGTTCGTGCCGTTATCGACCTTGAACGTGTTCATGGCGTCTCTGAA
AT5G04140.1_variant_of_AtGLU1 TCTCGTCTTTCCTCTGTTCGTGCTGTTATCGACCTTGAACGTCTTAATGGCGTCTCTCAC
*********************** ****************** ** *********** *
AT5G04140_-_AtGLU1 AAGGATCTCTCCTCTCCTTCAGCTTTAAGACCTCAGGTTCGTTTCTTCACTGATATAAAT
AT5G04140.1_variant_of_AtGLU1 AAAGATCTCTCTTCTACTTCACTCTTAAAACC----------------------------
** ******** *** ***** **** ***
AT5G04140_-_AtGLU1 TTCACAAACACCCAACGTGCAAAGTTTCATCCTTTATGGGGATCTTTTAAGCAGGTTGCT
AT5G04140.1_variant_of_AtGLU1 --------------------------------------------------TCAGGTTGCT
*********
AT5G04140_-_AtGLU1 AATTTGGAGGATATATTGTCTGAAAGAGGAGCTTGTGGAGTTGGGTTTATAGCAAACTTA
AT5G04140.1_variant_of_AtGLU1 AATTTGGAGGACATATTGTCTGAAAGAGGAGCTTGTGGAGTTGGGTTTATAGCAAACTTA
*********** ************************************************
AT5G04140_-_AtGLU1 GATAACATACCTTCACATGGAGTTGTCAAAGATGCTCTTATTGCTCTTGGGTGTATGGAA
AT5G04140.1_variant_of_AtGLU1 GATAACATACCTTCACATGGAGTTGTCAAAGATGCTCTTATTGCTCTTGGTTGTATGGAA
************************************************** *********
AT5G04140_-_AtGLU1 CATCGTGGAGGTTGTGGAGCAGACAATGATTCTGGTGATGGCTCTGGTCTTATGTCTTCC
AT5G04140.1_variant_of_AtGLU1 CATCGTGGAGGTTGTGGAGCTGACAATGATTCTGGTGATGGCTCTGGTCTTATGTCTTCC
******************** ***************************************
AT5G04140_-_AtGLU1 ATTCCTTGGGATTTCTTTAACGTCTGGGCCAAGGAACAAAGTCTTGCTCCTTTTGATAAG
AT5G04140.1_variant_of_AtGLU1 ATTCCTTGGGATTTCTTTAACGTCTGGGCCAAGGAACAAGGTCTTTCTCCTTTTGATAAG
*************************************** ***** **************
AT5G04140_-_AtGLU1 TTGCATACTGGTGTTGGCATGATCTTTCTTCCACAAGATGATACCTTTATGCAAGAAGCC
AT5G04140.1_variant_of_AtGLU1 TTGCATACTGGTGTTGGCATGATCTTTCTTCCACAAGAGGATACCTTTATGCAAGAAGCC
************************************** *********************
AT5G04140_-_AtGLU1 AAGCAAGTTATTGAAAACATATTTGAGAAAGAAGGATTACAAGTTCTTGGGTGGAGGGAA
AT5G04140.1_variant_of_AtGLU1 AAACAAGTTATTGAAAACATATTTGAGAAAGAAGGATTACAAGTTCTTGGTTGGAGGGAA
** *********************************************** *********
AT5G04140_-_AtGLU1 GTTCCTGTTAATGTTCCTATAGTTGGTAAAAATGCTAGGGAGACAATGCCTAACATTCAA
AT5G04140.1_variant_of_AtGLU1 GTTCCTGTGAATGTTCCTATAGTTGGTAAAAATGCTAGGGAAACAATGCCTAACATTCAA
******** ******************************** ******************
AT5G04140_-_AtGLU1 CAAGTGTTTGTGAAAATCGCAAAGGAAGATAGTACTGATGATATTGAAAGGGAACTTTAC
AT5G04140.1_variant_of_AtGLU1 CAAGTGTTTGTCAAAATCGCAAAGGAAGATAGTACGGATGACATTGAAAGGGAACTTTAC
*********** *********************** ***** ******************
AT5G04140_-_AtGLU1 ATCTGCCGGAAACTAATCGAAAGGGCCGTAGCTACTGAGAGTTGGGGAACAGAGCTTTAC
AT5G04140.1_variant_of_AtGLU1 ATATGCCGTAAACTAATCGAAAGGGCGGTGGCTACTGAGAGTTGGGGAACAGAGCTATAC
** ***** ***************** ** ************************** ***
AT5G04140_-_AtGLU1 TTCTGTTCACTGTCCAATCAAACCATAGTGTACAAGGGCATGCTTCGATCTGAAGCTCTT
AT5G04140.1_variant_of_AtGLU1 TTCTGTTCACTGTCCAATCAAACCATAGTGTACAAGGGCATGCTTCGGTCCGAGGCTCTT
*********************************************** ** ** ******
AT5G04140_-_AtGLU1 GGATTGTTTTATCTAGATCTTCAGAATGAGCTTTATGAGTCTCCTTTTGCTATTTATCAT
AT5G04140.1_variant_of_AtGLU1 GGATTGTTTTATCTAGATCTTCAGAATGAGCTTTATGAGTCTCCCTTTGCTATTTATCAT
******************************************** ***************
AT5G04140_-_AtGLU1 CGAAGGTACAGTACAAACACTAGTCCTAGGTGGCCTCTTGCTCAACCAATGAGGTTTCTT
AT5G04140.1_variant_of_AtGLU1 CGAAGGTACAGTACAAACACTAGTCCTAGGTGGCCTCTTGCTCAACCAATGAGGTTTCTT
************************************************************
AT5G04140_-_AtGLU1 GGACATAACGGGGAGATCAATACCATTCAGGGGAACTTAAATTGGATGCAGTCTCGAGAA
AT5G04140.1_variant_of_AtGLU1 GGACATAACGGAGAGATCAATACCATACAGGGGAACTTAAATTGGATGCAGTCTCGAGAA
*********** ************** *********************************
AT5G04140_-_AtGLU1 GCTTCATTGAAGGCTGCTGTTTGGAATGGCCGTGAAAATGAAATTCGTCCATTTGGTAAT
AT5G04140.1_variant_of_AtGLU1 GCTTCTTTGAAGTCCTCTGTTTGGAATGGCCGTGAAAATGAAATTCGTCCATTTGGTAAT
***** ****** * ********************************************
AT5G04140_-_AtGLU1 CCCAGGGGTTCAGACTCTGCTAATCTTGATAGTGCTGCAGAAATCATGATTAGAAGTGGA
AT5G04140.1_variant_of_AtGLU1 CCCAGAGGTTCAGACTCTGCTAATCTTGATAGTGCTGCAGAAATCTTGATTAGAAGTGGC
***** *************************************** *************
AT5G04140_-_AtGLU1 AGAACACCAGAGGAAGCTCTAATGATTCTTGTCCCTGAGGCATACAAGAATCATCCAACC
AT5G04140.1_variant_of_AtGLU1 CGAACTCCAGAGGAGGCTCTAATGATTCTTGTCCCTGAGGCATACAAAAATCATCCAACC
**** ******** ******************************** ************
AT5G04140_-_AtGLU1 TTATCTGTCAAATATCCTGAGGTTGTAGATTTCTATGACTACTACAAAGGACAAATGGAG
AT5G04140.1_variant_of_AtGLU1 TTATCTGTCAAATATCCTGAGGTTATAGATTTCTATGACTACTACAAAGGACAAATGGAG
************************ ***********************************
AT5G04140_-_AtGLU1 GCTTGGGATGGTCCTGCCTTACTTTTGTTCAGTGATGGAAAAACAGTTGGGGCTTGTCTT
AT5G04140.1_variant_of_AtGLU1 GCTTGGGATGGTCCTGCCTTACTTTTGTTCAGTGATGGAAAAACAGTTGGGGCTTGTCTT
************************************************************
AT5G04140_-_AtGLU1 GACCGTAATGGCCTTCGTCCTGCTCGATATTGGCGGACTAGTGACAATTTCGTCTATGTT
AT5G04140.1_variant_of_AtGLU1 GACCGTAATGGCCTTCGTCCTGCTCGATATTGGCGGACTAGTGACAATGTCGTCTATGTA
************************************************ **********
AT5G04140_-_AtGLU1 GCATCTGAGGTCGGAGTTGTACCAGTTGATGAGGCAAAAGTCACAATGAAAGGCCGTCTA
AT5G04140.1_variant_of_AtGLU1 GCATCTGAGGTCGGAGTTGTACCAGTTGATGAGGCAAAAGTCACGATGAAAGGCCGTCTA
******************************************** ***************
AT5G04140_-_AtGLU1 GGACCTGGAATGATGATTGCTGTTGACCTTGTGAATGGCCAGGTATATGAGAATACAGAG
AT5G04140.1_variant_of_AtGLU1 GGACCTGGAATGATGATTGCTGTTGACCTAGTGAATGGCCAGGTATATGAGAATACAGAG
***************************** ******************************
AT5G04140_-_AtGLU1 GTCAAGAAGAGAATATCTTCATTTAATCCATATGGAAAATGGATTAAAGAAAACTCCCGG
AT5G04140.1_variant_of_AtGLU1 GTCAAGAAGAGAATATCTTCATTTAATCCATATGGAAAATGGATTAAAGAAAACTCCCGG
************************************************************
AT5G04140_-_AtGLU1 TTCTTGAAGCCTGTGAATTTCAAATCCTCAACTGTCATGGAAAATGAAGAAATCCTAAGA
AT5G04140.1_variant_of_AtGLU1 TTCTTGAAGCCTGTGAATTTCAAATCCTCAACTGTCATGGAAAATGAAGAAATCTTAAGA
****************************************************** *****
AT5G04140_-_AtGLU1 AGCCAACAAGCATTTGGTTATTCAAGTGAGGATGTGCAAATGGTTATTGAGTCTATGGCT
AT5G04140.1_variant_of_AtGLU1 AGCCAACAGGCATTTGGTTATTCAAGTGAGGATGTGCAAATGGTTATTGAGTCTATGGCT
******** ***************************************************
AT5G04140_-_AtGLU1 TCTCAAGGAAAGGAACCAACCTTCTGCATGGGCGACGATATTCCGCTGGCAGGATTGTCT
AT5G04140.1_variant_of_AtGLU1 TCTCAAGGAAAGGAACCAACCTTCTGCATGGGCGACGATATTCCGCTGGCAGGATTGTCT
************************************************************
AT5G04140_-_AtGLU1 CAAAGACCGCATATGCTTTATGATTATTTCAAACAAAGATTTGCACAGGTTACAAACCCT
AT5G04140.1_variant_of_AtGLU1 CAAAGACCACATATGCTTTACGATTATTTCAAACAAAGATTTGCGCAGGTTACAAACCCT
******** *********** *********************** ***************
AT5G04140_-_AtGLU1 GCCATTGACCCCCTTAGGGAAGGTTTGGTTATGTCTCTTGAAGTAAATATTGGAAAACGT
AT5G04140.1_variant_of_AtGLU1 GCCATTGATCCCCTTAGGGAAGGTTTGGTTATGTCTCTTGAAGTAAATATTGGAAAACGT
******** ***************************************************
AT5G04140_-_AtGLU1 GGAAATATATTGGAGCTTGGACCTGAGAATGCCTCGCAGGTTATTCTGTCTAACCCTGTG
AT5G04140.1_variant_of_AtGLU1 GGAAATATATTGGAGCTTGGGCCTGAGAATGCCTCACAGGTTATTCTGTCTAACCCTGTG
******************** ************** ************************
AT5G04140_-_AtGLU1 TTAAATGAAGGAGCGTTAGAGGAGTTAATGAAGGATCAATACTTAAAACCAAAGGTTCTG
AT5G04140.1_variant_of_AtGLU1 TTAAACGAAGGAACGTTGGAGGAGTTAATGAAGGATACATACTTAAAACCTAAGGTTCTG
***** ****** **** ****************** ************ *********
AT5G04140_-_AtGLU1 TCCACATATTTCGATATAAGAAAAGGCGTTGAAGGTTCCTTGCAAAAGGCTCTATATTAT
AT5G04140.1_variant_of_AtGLU1 TCCACATATTTCGATATACGAAAAGGCGTTGAAGGTTCCTTGCAAAAGGCTCTATATTCT
****************** *************************************** *
AT5G04140_-_AtGLU1 CTTTGTGAAGCAGCTGATGATGCTGTCCGAAGTGGCTCTCAGCTTCTCGTTCTTTCAGAC
AT5G04140.1_variant_of_AtGLU1 CTTTGTGAAGCAGCTGATGATGCTGTCCGAAGTGGCTCTCAGCTTCTCGTTCTTTCAGAC
************************************************************
AT5G04140_-_AtGLU1 CGATCCGATAGACTGGAACCAACCAGGCCTTCAATTCCAATAATGTTAGCTGTTGGCGCT
AT5G04140.1_variant_of_AtGLU1 CGATCCGATAGCCTGGAACCAACCCGCCCTGCAATTCCGATAATGTTAGCTGTTGGCGCT
*********** ************ * *** ******* *********************
AT5G04140_-_AtGLU1 GTCCATCAACATCTTATTCAGAACGGCTTGCGTATGTCAGCTTCTATTGTTGCTGATACC
AT5G04140.1_variant_of_AtGLU1 GTCCATCAACATCTTATTCAGAACGTATCTTTTGTTTGCTCTGAATT-------------
************************* * * * * ** *
AT5G04140_-_AtGLU1 GCCCAATGCTTCAGCACACATCATTTTGCTTGTTTGGTTGGATATGGTGCAAGTGCTGTA
AT5G04140.1_variant_of_AtGLU1 --------------------------------TTCTAAATTTGTCCCTGCCAATGCTGTA
** *** * *******
AT5G04140_-_AtGLU1 TGCCCATACTTGGCACTGGAGACATGTAGGCAATGGCGCTTAAGTAACAAAACTGTGGCC
AT5G04140.1_variant_of_AtGLU1 TGCCCATACTTGGCACTGGAGACATGTAGGCAATGGCGCTTAAGTAACAAAACTGTCGCA
******************************************************** **
AT5G04140_-_AtGLU1 TTCATGCGTAACGGGAAAATTCCTACTGTAACCATTGAGCAAGCTCAGAAGAACTACACT
AT5G04140.1_variant_of_AtGLU1 TTCATGCGTAACGGGAAAATTCCCACTGTAACCATTGAGCAAGCTCAGAAAAACTACACC
*********************** ************************** ********
AT5G04140_-_AtGLU1 AAGGCGGTTAATGCAGGGCTTCTTAAAATTCTTTCTAAGATGGGAATCTCATTGCTTTCA
AT5G04140.1_variant_of_AtGLU1 AAGGCGGTTAATGCAGGGCTTCTTAAAATTCTTTCTAAGATGGGCATCTCATTGCTTTCA
******************************************** ***************
AT5G04140_-_AtGLU1 AGTTATTGTGGTGCTCAGATATTTGAGATATATGGTTTGGGACAGGATGTTGTTGATCTT
AT5G04140.1_variant_of_AtGLU1 AGTTATTGTGGTGCTCAGATATTTGAGATCTATGGTTTGGGGAAAGAGGTTGTTGATCTT
***************************** *********** * ** ************
AT5G04140_-_AtGLU1 GCATTCACTGGAAGTGTGTCAAAAATCAGTGGACTCACCTTTGATGAGTTGGCAAGAGAG
AT5G04140.1_variant_of_AtGLU1 GCATTCACTGGAAGTGTGTCAAAAATCAGTGGACTCACCTTTGATGAGTTGGCAAGAGAG
************************************************************
AT5G04140_-_AtGLU1 ACATTGTCTTTCTGGGTGAAGGCCTTTTCTGAGGATACAACTAAGCGATTAGAAAATTTT
AT5G04140.1_variant_of_AtGLU1 ACATTGTCTTTTTGGGTGAAGGCCTTTTCTGAGGATACAACTAAACGATTAGAAAATTTC
*********** ******************************** **************
AT5G04140_-_AtGLU1 GGGTTTATTCAATTCAGGCCTGGAGGTGAGTATCATTCAAACAACCCAGAGATGTCAAAG
AT5G04140.1_variant_of_AtGLU1 GGGTTTATTCAATTTAGGCCTGGAGGTGAGTATCATTCAAACAACCCAGAGATGTCAAAG
************** *********************************************
AT5G04140_-_AtGLU1 TTGCTTCACAAGGCTGTCCGTGAAAAGAGTGAAACTGCATATGCAGTCTATCAACAGCAT
AT5G04140.1_variant_of_AtGLU1 TTGCTCCACAAGGCTGTCCGTGAAAAGAGTGAAACTGCATATGCAGTCTATCAACAACAT
***** ************************************************** ***
AT5G04140_-_AtGLU1 CTCTCTAACAGACCTGTTAATGTCCTCCGTGACCTGCTTGAGTTCAAGAGTGATCGTGCA
AT5G04140.1_variant_of_AtGLU1 CTCTCTAACAGACCTGTTAATGTCCTCCGTGATCTCCTTGAGTTCAAGAGTGACCGTGCA
******************************** ** ***************** ******
AT5G04140_-_AtGLU1 CCGATCCCAGTAGGGAAAGTAGAACCGGCCGTTGCTATTGTTCAGAGATTTTGTACTGGT
AT5G04140.1_variant_of_AtGLU1 CCGATCCCAGTAGGGAAAGTTGAACCAGCCGTTTCCATTGTCCAGAGATTTTGTACTGGT
******************** ***** ****** * ***** ******************
AT5G04140_-_AtGLU1 GGAATGTCACTTGGTGCTATTTCAAGAGAGACTCATGAAGCTATTGCTATTGCAATGAAT
AT5G04140.1_variant_of_AtGLU1 GGAATGTCACTTGGTGCTATTTCAAGAGAGACTCATGAAGCTATTGCTATTGCTATGAAT
***************************************************** ******
AT5G04140_-_AtGLU1 AGGATTGGTGGGAAATCAAACTCTGGAGAAGGTGGAGAGGATCCTATCCGTTGGAAGCCA
AT5G04140.1_variant_of_AtGLU1 AGGATTGGTGGTAAATCAAACTCTGGAGAAGGTGGAGAGGATCCTATCCGTTGGAAGCCA
*********** ************************************************
AT5G04140_-_AtGLU1 CTTACAGATGTGGTTGATGGATATTCACCAACACTACCACATCTCAAAGGTCTTCAAAAC
AT5G04140.1_variant_of_AtGLU1 CTTACAGATGTGGTTGATGGGTATTCATCAACATTACCACATCTCAAAGGTCTTCAAAAC
******************** ****** ***** **************************
AT5G04140_-_AtGLU1 GGCGATATTGCAACAAGTGCTATCAAGCAGGTTGCTTCAGGGCGTTTTGGAGTCACACCA
AT5G04140.1_variant_of_AtGLU1 GGTGATATCGCAACAAGTGCTATCAAGCAGGTTGCTTCAGGACGTTTTGGAGTCACACCA
** ***** ******************************** ******************
AT5G04140_-_AtGLU1 ACGTTCTTGGTCAATGCAGATCAATTGGAAATCAAAGTTGCACAAGGTGCCAAGCCTGGG
AT5G04140.1_variant_of_AtGLU1 ACATTCTTGGTCAATGCAGACCAATTGGAAATCAAAGTTGCACAAGGTGCCAAGCCTGGG
** ***************** ***************************************
AT5G04140_-_AtGLU1 GAAGGTGGTCAGCTTCCTGGAAAGAAAGTTAGTGCGTATATCGCTAGGCTAAGAAGCTCT
AT5G04140.1_variant_of_AtGLU1 GAAGGTGGTCAGCTTCCTGGAAAGAAAGTTAGTGCATATATCGCAAGGCTAAGAAGCTCT
*********************************** ******** ***************
AT5G04140_-_AtGLU1 AAACCTGGTGTTCCGCTTATATCTCCGCCTCCTCACCACGACATTTACTCTATTGAGGAT
AT5G04140.1_variant_of_AtGLU1 AAACCTGGTGTTCCGCTTATATCCCCGCCTCCTCACCACGACATTTACTCTATTGAGGAT
*********************** ************************************
AT5G04140_-_AtGLU1 CTTGCTCAGTTGATCTTTGATCTACATCAGATTAATCCAAATGCAAAAGTATCAGTCAAG
AT5G04140.1_variant_of_AtGLU1 CTTGCTCAGTTGATCTTTGATCTACATCAGATTAATCCTAACGCAAAAGTATCAGTCAAG
************************************** ** ******************
AT5G04140_-_AtGLU1 CTAGTTGCAGAAGCTGGAATCGGAACTGTTGCTTCAGGAGTTGCAAAGGGTAACGCTGAT
AT5G04140.1_variant_of_AtGLU1 CTGGTCGCAGAAGCTGGAATCGGAACAGTTGCTTCTGGAGTTGCAAAGGGTAACGCTGAT
** ** ******************** ******** ************************
AT5G04140_-_AtGLU1 ATCATCCAGATATCAGGCCATGATGGTGGAACCGGGGCTAGTCCAATAAGCTCCATAAAA
AT5G04140.1_variant_of_AtGLU1 ATCATTCAGATATCAGGGCATGATGGTGGAACTGGTGCTAGTCCAATAAGCTCCATAAAA
***** *********** ************** ** ************************
AT5G04140_-_AtGLU1 CATGCTGGTGGACCATGGGAACTTGGACTAACAGAAACTCACCAAACACTTATCGCAAAT
AT5G04140.1_variant_of_AtGLU1 CATGCTGGTGGACCATGGGAACTTGGACTAACAGAAACTCACCAAACTCTTATCGAAAAT
*********************************************** ******* ****
AT5G04140_-_AtGLU1 GGACTAAGAGAAAGAGTCATTTTAAGAGTCGATGGAGGCTTAAAGAGTGGTGTTGATGTT
AT5G04140.1_variant_of_AtGLU1 GGACTCAGAGAAAGAGTCATTTTAAGAGTCGATGGAGGCTTAAAGAGTGGTGTTGATGTT
***** ******************************************************
AT5G04140_-_AtGLU1 CTAATGGCTGCAGCTATGGGTGCTGATGAATACGGATTCGGTTCCTTGGCAATGATTGCT
AT5G04140.1_variant_of_AtGLU1 CTAATGGCTGCAGCTATGGGTGCTGATGAATACGGATTTGGTTCCTTGGCAATGATTGCT
************************************** *********************
AT5G04140_-_AtGLU1 ACTGGTTGTGTTATGGCTCGTATTTGCCACACTAATAATTGCCCAGTGGGTGTAGCAAGT
AT5G04140.1_variant_of_AtGLU1 ACTGGTTGTGTTATGGCTCGTATTTGCCACACTAATAACTGCCCAGTGGGTGTAGCAAGT
************************************** *********************
AT5G04140_-_AtGLU1 CAGAGAGAAGAATTACGTGCAAGATTCCCTGGTGTACCTGGTGATCTTGTCAACTACTTC
AT5G04140.1_variant_of_AtGLU1 CAGAGAGAAGAATTACGTGCAAGATTCCCCGGTGTGCCTGGTGATCTTGTCAATTACTTC
***************************** ***** ***************** ******
AT5G04140_-_AtGLU1 TTATACGTAGCAGAAGAGGTGAGAGGTATCTTAGCACAGTTGGGATACAACAGTTTAGAT
AT5G04140.1_variant_of_AtGLU1 TTATACGTAGCAGAAGAGGTGAGAGGTATCTTAGCACAGTTGGGATACAGCAAGTTAGAT
************************************************* ** ******
AT5G04140_-_AtGLU1 GACATCATTGGACGAACAGAGTTACTGAGACCACGAGACATTTCGCTAGTTAAAACTCAA
AT5G04140.1_variant_of_AtGLU1 GACATCATTGGACGAACAGAGTTACTGAAACCACGAGACATTTCGTTAGTGAAAACTCAG
**************************** **************** **** ********
AT5G04140_-_AtGLU1 CATCTCGATCTGAGTTATCTTCTTTCGTCTGTTGGAACACCTTCATTGAGCAGTACTGAA
AT5G04140.1_variant_of_AtGLU1 CATCTCGATCTGAGTTATCTTCTTTCGTCTGTTGGAACACCTTCATTGAGCAGTACTGAA
************************************************************
AT5G04140_-_AtGLU1 ATCAGAAAGCAGGAAGTTCATACAAATGGACCTGTTCTCGACGACGATATTCTTGCAGAT
AT5G04140.1_variant_of_AtGLU1 ATCAGAAAGCAGGAAGTTCATACAAATGGACCTGTTCTCGACGACGATATTCTTGCAGAT
************************************************************
AT5G04140_-_AtGLU1 CCATTGGTGATTGATGCAATAGAGAACGAAAAAGTGGTTGAGAAAACCGTCAAAATATGC
AT5G04140.1_variant_of_AtGLU1 CCATTGGTGATTGATGCAATAGAGAACGAAAAGGTGGTTGATAAAACCGTCAAAATATGC
******************************** ******** ******************
AT5G04140_-_AtGLU1 AACGTAGACCGTGCGGCTTGTGGTCGTGTTGCTGGTGTTATTGCAAAGAAGTATGGAGAC
AT5G04140.1_variant_of_AtGLU1 AACGTAGACCGTGCCGTTTGTGGTCGCGTTGCTGGTGTTATTGCAAAGAAGTATGGAGAC
************** * ********* *********************************
AT5G04140_-_AtGLU1 ACTGGTTTTGCAGGACAAGTGAACCTAACTTTCTTAGGGAGCGCGGGACAGTCGTTTGGG
AT5G04140.1_variant_of_AtGLU1 ACTGGTTTTGCAGGACAAGTGAATCTGACTTTCCTTGGGAGCGCGGGACAGTCCTTTGGG
*********************** ** ****** * ***************** ******
AT5G04140_-_AtGLU1 TGCTTTTTGATTCCCGGTATGAACATCCGGCTCATAGGAGAGTCAAATGACTACGTTGGA
AT5G04140.1_variant_of_AtGLU1 TGCTTTTTGATTCCCGGTATGAACATCCGGCTAATAGGAGAGTCAAATGACTACGTTGGA
******************************** ***************************
AT5G04140_-_AtGLU1 AAGGGAATGGCTGGTGGTGAAATAGTAGTAACTCCTGTGGAAAAAATCGGGTTTGTGCCT
AT5G04140.1_variant_of_AtGLU1 AAGGGAATGGCTGGTGGTGAAATAGTAGTAACTCCTGTGGACACAATCGGGTTTGTGCCC
***************************************** * ***************
AT5G04140_-_AtGLU1 GAGGAAGCAACGATAGTCGGGAACACTTGCTTGTATGGTGCAACAGGTGGTCAGATATTC
AT5G04140.1_variant_of_AtGLU1 GAGGAAGCAACGATAGTCGGGAACACTTGCTTGTATGGTGCCACAGGCGGTCAGATATTT
***************************************** ***** ***********
AT5G04140_-_AtGLU1 GCTAGAGGCAAAGCTGGAGAGAGATTTGCAGTGAGAAACTCACTCGCTGAAGCAGTAGTT
AT5G04140.1_variant_of_AtGLU1 GCTAGAGGCAAAGCTGGAGAGAGATTTGCAGTGAGAAACTCACTAGCTGAAGCAGTAGTT
******************************************** ***************
AT5G04140_-_AtGLU1 GAAGGCACTGGAGACCATTGCTGTGAGTACATGACTGGTGGCTGTGTAGTCGTGCTTGGA
AT5G04140.1_variant_of_AtGLU1 GAAGGCACCGGAGACCATTGCTGTGAGTACATGACTGGTGGCTGTGTCGTCGTGCTTGGA
******** ************************************** ************
AT5G04140_-_AtGLU1 AAGGTGGGAAGAAACGTTGCTGCTGGTATGACAGGAGGGTTAGCTTACCTTCTTGATGAA
AT5G04140.1_variant_of_AtGLU1 AAGGTGGGAAGAAACGTTGCTGCGGGTATGACTGGAGGCTTAGCTTACCTTCTTGATGAA
*********************** ******** ***** *********************
AT5G04140_-_AtGLU1 GACGACACTCTTCTTCCTAAGATTAACAGAGAGATAGTGAAGATCCAAAGAGTAACTGCG
AT5G04140.1_variant_of_AtGLU1 GATGACACTCTTCTCCCTAAGATTAACCGAGAGATAGTGAAGATGCAAAGAGTAACTGCG
** *********** ************ **************** ***************
AT5G04140_-_AtGLU1 CCTGCAGGGGAATTGCAGCTGAAGAGCTTAATTGAAGCACATGTGGAAAAAACCGGAAGC
AT5G04140.1_variant_of_AtGLU1 CCTGCAGGGGAATTGCAGCTGAAGAGCTTAATTGAAGCACATGTCGAAAAAACCGGAAGC
******************************************** ***************
AT5G04140_-_AtGLU1 AGCAAAGGCGCAACGATTCTGAATGAGTGGGAAAAGTATCTACCTCTCTTCTGGCAACTG
AT5G04140.1_variant_of_AtGLU1 AGCAAAGGCGCAACGATTCTGAATGAATGGGACAAGTATCTACCTCTCTTCTGGCAACTG
************************** ***** ***************************
AT5G04140_-_AtGLU1 GTTCCACCGAGTGAGGAAGACACTCCTGAAGCTTCTGCTGCTTACGTAAGAACATCCACC
AT5G04140.1_variant_of_AtGLU1 GTTCCACCGAGTGAAGAAGACACTCCTGAAGCTTCTGCTGCATATGTAAGAACAGCCACC
************** ************************** ** ********* *****
AT5G04140_-_AtGLU1 GGGGAAGTCACATTTCAATCGGCTTAG
AT5G04140.1_variant_of_AtGLU1 GGAGAAGTCACATTTCAATCGGCATAA
** ******************** **
CLUSTAL O(1.2.1) multiple sequence alignment of AtGLU2 (AT2G41220) and its splicing variant
|
Gene |
Locus |
5'-Forwad primer -3' |
5'-Reverse primer -3' |
GC content (%) F/R |
Amplicon size (bp) |
|
AtGLU2 |
AT2G41220 |
GCAGCATTTAGCCAACCGTC |
AGGCTCAACCTTCCCAACAG |
55.0/55.0 |
94 |
AT2G41220 ATGGCTCTACAGTCTCCCGGAGCTACCGGAGCTTCATCTTCCGTTTCCCGGCTTCTCTCT
GLU2_VARIANT_AT2G41220.1.1 ATGGCGCTACAGTCTCCCGGCGCCGCCGGAGCTTCGTCTTCCGTTTCCCGGCTTCTAACC
***** ************** ** ********** ******************** *
AT2G41220 TCCGCGAAATTAAGCTCTACTAAGACTATCTTCTCTGTTGACTTCGTCAGATCCTACTGT
GLU2_VARIANT_AT2G41220.1.1 TCCGCGAAATTAAGCTCTACTAAGACTATCTTCTCTGTTGACTTCGTCGGATCCTACTGT
************************************************ ***********
AT2G41220 ATTTCTAAAGGAACCAAACGGCGTAACGAACTCTCCGGCTTTCGCGGCTACAGTCCACTG
GLU2_VARIANT_AT2G41220.1.1 ATTTCTAAAGGAACTAAACGGCGTAACGAACTCTCCGGGTTTCGCGGCTACAGTCCACTG
************** *********************** *********************
AT2G41220 CTCAAGTCCTCGCTGAGGTCTCCGTTTTCGGTGAAAGCGATCCTTAATTCTGACCGAGCC
GLU2_VARIANT_AT2G41220.1.1 CTCAAGTCCTCGCTGAGGTCTCCGTTTTCGGCCAAAGCGATCCTGAATTCTGACCGGGCC
******************************* *********** *********** ***
AT2G41220 GCTGGTGATGCATCGTCTTCCTTTTCTGATCTGAAACCTCAGGTGGCTTATTTGGAAGAT
GLU2_VARIANT_AT2G41220.1.1 GCTGGTGATGCATCGGCTTCCTTTTCTGATCTGAAACCTCAGGTGGCCTACTTGGAAGAT
*************** ******************************* ** *********
AT2G41220 ATAATATCGGAAAGGGGAGCATGTGGTGTTGGGTTTATTGCAAACTTGGAGAATAAGGCT
GLU2_VARIANT_AT2G41220.1.1 ATAATTTCAGAAAGGGGAGCATGTGGTGTTGGGTTTATCGCGAACTTGGAGAATAAGGCT
***** ** ***************************** ** ******************
AT2G41220 ACACATAAGATTGTAAATGATGCTCTTATAGCACTTGGTTGTATGGAACACAGGGGAGGT
GLU2_VARIANT_AT2G41220.1.1 ACACATAAGATTGTAAATGATGCTCTCATAGCACTTGGTTGTATGGAACACAGGGGAGGT
************************** *********************************
AT2G41220 TGTGGTTCTGATAATACTTCTGGTGATGGTTCTGGTTTGATGACTTCCATTCCTTGGGAT
GLU2_VARIANT_AT2G41220.1.1 TGTGGTTCTGATAATACTTCTGGTGATGGTTCTGGTTTGATGACTTCCATTCCTTGGGAT
************************************************************
AT2G41220 CTATTTAACGAATGGGCTGAGAAGCAAGGAATCGCTTCTTTTGATAGGACGCATACGGGT
GLU2_VARIANT_AT2G41220.1.1 CTATTTAACGAATGGGCTGAGAAGCAAGGAATGGCTTCTTTTGATAAGACGCATACGGGT
******************************** ************* *************
AT2G41220 GTTGGAATGCTGTTCCTACCGAGAGATGACAACATTAGGAAAGAAGCCAAGAAAGTGATT
GLU2_VARIANT_AT2G41220.1.1 GTTGGAATGCTCTTCCTACCGAGAGATGACAACATTAGGGCAGAAGCCAAGAAAGTGATT
*********** *************************** *******************
AT2G41220 ACAAGTATTTTTGAGAAAGAGGGCCTGGAGGTGCTTGGATGGAGAGATGTTCCTGTGGAA
GLU2_VARIANT_AT2G41220.1.1 ACAAGTATTTTTGAGAAAGAGGGCCTGGAGGTGCTTGGATGGAGAGATGTTCCTGTAGAA
******************************************************** ***
AT2G41220 GCGTCCATTGTTGGCCATAATGCAAAACAGACGATGCCAAACACGGAACAGGTCTTTGTT
GLU2_VARIANT_AT2G41220.1.1 GCGTCCATTGTCGGTCATAATGCAAAACAGACAATGCCAAACACGGAACAGGTCTTTGTT
*********** ** ***************** ***************************
AT2G41220 AGAATTGTCAAAGACGATAAAGTAGACGACGTTGAACGAGAACTTTACATTTGCAGGAAG
GLU2_VARIANT_AT2G41220.1.1 AGAATTGTCAAAGACGATAAAGTAGACGACGTTGAACGAGAACTTTACATTTGCAGGAAG
************************************************************
AT2G41220 CTGATCGAAAGAGCAGTTGCTTCTGAAAGTTGGGCCTCTGAGCTCTACTTTAGCTCCTTG
GLU2_VARIANT_AT2G41220.1.1 CTGATCGAAAGAGCAGTTGCTTCTGAAAGTTGGGCTTCTGAGCTCTACTTTAGCTCCTTA
*********************************** ***********************
AT2G41220 TCCAATCAGACCATTGTTTACAAGGGAATGCTTCGTTCTGAAGTTCTTGGGTTATTTTAC
GLU2_VARIANT_AT2G41220.1.1 TCCAATCAGACCATTGTTTACAAGGGAATGCTTCGTTCTGAAGTTCTTGGGTTATTTTAC
************************************************************
AT2G41220 CCCGACCTTCAAAATGATCTTTATAAATCTCCTTTCGCCATCTATCATCGAAGATTTAGC
GLU2_VARIANT_AT2G41220.1.1 CCCGACCTTCAAAATGATCTTTATAAATCTGCTTTTGCCATCTATCATCGAAGATTTAGC
****************************** **** ************************
AT2G41220 ACAAATACTAGTCCCAGATGGCATCTTGCTCAGCCCATGAGGTTTCTTGGGCACAATGGG
GLU2_VARIANT_AT2G41220.1.1 ACAAATACTAGTCCCAGATGGCATCTTGCTCAACCCATGAGGTTTCTTGGACACAATGGG
******************************** ***************** *********
AT2G41220 GAGATCAATACCATACAGGGGAACCTAAACTGGATGACGTCTCGAGAAGCATCTCTAAGA
GLU2_VARIANT_AT2G41220.1.1 GAGATCAATACCATACAGGGGAACCTAAACTGGATGACTTCACGAGAAGCATCTCTAAGA
************************************** ** ******************
AT2G41220 TCACCTGTCTGGCATGGACGTGAAAATGACATTCGTCCAATTAGCAATCCAAAGGCTTCA
GLU2_VARIANT_AT2G41220.1.1 TCACCTGTCTGGCATGGACGTGAAAATGATATTCGTCCAATTAGCAATCCAAAGGCTTCA
***************************** ******************************
AT2G41220 GACTCTGCTAACCTTGACAGTGCGGCGGAGCTATTGATAAGAAGTGGCCGGACTCCAGAG
GLU2_VARIANT_AT2G41220.1.1 GACTCTGCTAACCTTGACAGTGCAGCGGAGCTATTGATAAGAAGTGGCCGGACTCCAGAG
*********************** ************************************
AT2G41220 GAATCTCTTATGATTCTTGTCCCAGAGGCGTATAAAAATCACCCGACTTTAATGATTAAA
GLU2_VARIANT_AT2G41220.1.1 GAATCTCTTATGATTCTTGTCCCAGAGGCGTATAAAAATCACCCGACTTTAATGATTAAA
************************************************************
AT2G41220 TACCCTGAGGCTGTAGACTTCTATGACTATTACAAGGGGCAGATGGAGCCCTGGGATGGA
GLU2_VARIANT_AT2G41220.1.1 TACCCTGAGGCTGTAGACTTCTATGATTATTACAAGGGCCAAATGGAGCCCTGGGATGGA
************************** *********** ** ******************
AT2G41220 CCTGCTTTGGTATTGTTCAGTGATGGAAAGACTGTTGGAGCTTGCCTTGACCGCAATGGA
GLU2_VARIANT_AT2G41220.1.1 CCTGCTTTGGTATTGTTCAGTGATGGAAAGACTGTTGGAGCTTGCCTTGACCGCAATGGA
************************************************************
AT2G41220 CTTCGACCAGCTAGATATTGGCGGACAAGTGATAATGTTGTCTATGTAGCCTCTGAGGTC
GLU2_VARIANT_AT2G41220.1.1 CTTCGACCTGCTAGATATTGGCGGACAAGTGATAATGTTGTCTATGTAGCCTCTGAGGTC
******** ***************************************************
AT2G41220 GGAGTTCTTCCAATGGATGAATCGAAAGTCACCATGAAGGGTCGTCTAGGACCTGGTATG
GLU2_VARIANT_AT2G41220.1.1 GGAGTTCTTCCAATGGATGAATCGAAAGTCACCATGAAGGGTCGGCTAGGACCTGGCATG
******************************************** *********** ***
AT2G41220 ATGATATCTGTTGACTTAGAGAATGGACAGGTATATGAGAACACAGAAGTGAAGAAGCGG
GLU2_VARIANT_AT2G41220.1.1 ATGATATCTGTTGACTTAGAGAGTGGACAGGTATATGAGAACACAGAAGTAAAGAGGCGG
********************** *************************** **** ****
AT2G41220 GTTGCATCATATAACCCATATGGAAAGTGGGTTAGTGAAAACCTGCGAAACCTGAAGCCT
GLU2_VARIANT_AT2G41220.1.1 GTTGCATCATATAACCCATATGGAAAGTGGGTCAGTAAAAACCTGCGAAACCTGAAGCCT
******************************** *** ***********************
AT2G41220 TCTAATTATCTCTCGTCAGCGATCCTGGAGACTGATGAGACCTTAAGACGCCAACAGGCG
GLU2_VARIANT_AT2G41220.1.1 TCTAATTTTCTCTCGTCAGCAATCATGGAGACTGATGAGACCTTAAGACGCCAACAGGCG
******* ************ *** ***********************************
AT2G41220 TTTGGCTACTCGAGTGAAGATGTTCAAATGGTAATTGAGTCGATGGCTGCACAAGGAAAA
GLU2_VARIANT_AT2G41220.1.1 TTTGGCTACTCAAGTGAAGATGTTCAAATGGTAATTGAGTCGATGGCTGCACAAGGAAAA
*********** ************************************************
AT2G41220 GAACCAACGTTTTGCATGGGGGATGATACTCCAGTGGCAGTATTGTCTCAAAAGCCACAT
GLU2_VARIANT_AT2G41220.1.1 GAACCGACGTTTTGCATGGGGGATGATACTCCAGTGGCAGTATTGTCTCAAAAGCCGCAT
***** ************************************************** ***
AT2G41220 ATGCTTTATGATTATTTCAAGCAGCGGTTTGCTCAGGTTACAAATCCAGCTATTGACCCT
GLU2_VARIANT_AT2G41220.1.1 ATGCTTTATGATTATTTCAAGCAGCGGTTTGCTCAGGTTACAAATCCAGCTATTGATCCT
******************************************************** ***
AT2G41220 CTCCGGGAAGGATTAGTCATGTCCCTTGAAGTTAATATTGGAAAGCGTGGAAATATATTG
GLU2_VARIANT_AT2G41220.1.1 CTTCGAGAAGGATTGGTCATGTCGCTTGAAGTTAATATTGGAAAGCGTGGAAATATATTG
** ** ******** ******** ************************************
AT2G41220 GAGGTTGGGCCTCAGAATGTTTCACAGGTTGTTTTGTCTGGTCCTGTGCTAAATGAACGG
GLU2_VARIANT_AT2G41220.1.1 GAGGTTGGACCTCAGAATGTTTCACAGGTTGTTTTGTCTGGTCCTGTACTAAATGAACGT
******** ************************************** ***********
AT2G41220 GAGCTTGAGGGTTTGCTTGGTGATCCACTGTTAAAATCTCAAATCTTGCCCACATTTTTT
GLU2_VARIANT_AT2G41220.1.1 GAGCTTGAGGGTTTGTTCAGTGATCCACAGTTAAAATCTCAAGTCTTGCCCACATTTTTT
*************** * ********* ************* *****************
AT2G41220 GACATTCGTAGAGGAATTGAAGGATCTTTGAAGAAGGGTCTTCTAAAACTTTGTGAAGCT
GLU2_VARIANT_AT2G41220.1.1 GACATTCATAGAGGAATTGAAGGATCTTTGAAGAAGGGTCTTCTAAAACTCTGTGAAGCT
******* ****************************************** *********
AT2G41220 GCAGATGAAGCTGTTCGTAATGGTTCCCAAGTACTTGTTCTCTCAGACAGATCTGATAAT
GLU2_VARIANT_AT2G41220.1.1 GCAGATGAAGCTGTTCGTAGTGGTTCCCAAGTACTTGTTCTCTCAGACAGATCTGATAAT
******************* ****************************************
AT2G41220 CCGGAACCAACTCGACCTGCAATTCCAATGTTGTTGGCTGTGGGTGCTGTTCACCAACAT
GLU2_VARIANT_AT2G41220.1.1 CCGGAACCAACTCGACCAGCAATTCCAATGTTGTTGGCTGTGGGTGCTGTTCACCAACAT
***************** ******************************************
AT2G41220 CTGATCCAGAACGGTCTTCGGATGTCAGCTTCGATAATTGCAGATACAGCTCAGTGCTTT
GLU2_VARIANT_AT2G41220.1.1 CTGATCCAGAATGGTCTTCGAATGTCAGCTTCGATAATTGCAGATACAGCTCAGTGCTTT
*********** ******** ***************************************
AT2G41220 AGTACACATCACTTTGCCTGTTTGATTGGATATGGAGCAAGTGCTATATGCCCGCACCTA
GLU2_VARIANT_AT2G41220.1.1 AGTACACATCACTTTGCCTGTTTGATTGGATATGGAGCAAGTGCTATATGCCCACACCTA
***************************************************** ******
AT2G41220 GCACTGGAGACGTGTAGACAGTGGCGTCTGAGCAACAAGACCGTGAACATGATGAGAAAC
GLU2_VARIANT_AT2G41220.1.1 GCATTGGAGACATGTAGACAGTGGCGTCTGAGCAACAAGACTGTGAACATGATGAGAAAC
*** ******* ***************************** ******************
AT2G41220 GGAAAAATGCCTACTGTAACTATGGAACAGGCTCAGAAGAATTACCGCAAGGCTGTTAAC
GLU2_VARIANT_AT2G41220.1.1 GGAAAAATGCCTACTGTAACTATGGAACAGGCTCAGAAGAATTACCGCAAGGCTGTTAAC
************************************************************
AT2G41220 ACTGGTCTTCTTAAGGTTCTCTCTAAGATGGGCATCTCGTTATTCTCTAGTTATTGTGGA
GLU2_VARIANT_AT2G41220.1.1 ACTGGTCTTCTAAAGGTTCTCTCGAAGATGGGCATCTCGTTATTCTCTAGTTATTGTGGA
*********** *********** ************************************
AT2G41220 GCACAAATATTTGAAATTTATGGATTGGGAAATGAGGTTGTGGAATTTTCGTTCCGTGGT
GLU2_VARIANT_AT2G41220.1.1 GCACAAATATTTGAAATTTATGGATTGGGAAATGAGGTTGTTGAATTTTCGTTCCGTGGT
***************************************** ******************
AT2G41220 AGTGCATCTCAAATTGGTGGATTAACTTTAGATGAGCTGGCTAGGGAGACGTTAACTTTC
GLU2_VARIANT_AT2G41220.1.1 AGTGCATCTCAAATTGGTGGATTAACTTTGGATGAGCTGGCTAGAGAGACATTAACTTTC
***************************** ************** ***** *********
AT2G41220 TGGGTGAGGGCATTTTCTGAGGATACAGCCAAACGGCTAGAGAACTTCGGTTTCATTCAG
GLU2_VARIANT_AT2G41220.1.1 TGGGTGAGGGCATTTTCTGAGGATACAGCCAAGCGGCTAGAGAACTTCGGTTTCATTCAG
******************************** ***************************
AT2G41220 TTTAGGCCTGGAGGTGAATATCATGGAAACAACCCAGAAATGTCTAAATTACTTCATAAA
GLU2_VARIANT_AT2G41220.1.1 TTTAGGCCTGGAGGCGAATATCATGGGAACAACCCAGAAATGTCTAAATTACTTCATAAA
************** *********** *********************************
AT2G41220 GCCGTCCGCGAAAAGAGTGAGACAGCCTATGCAGTCTATCAGCAGCATTTAGCCAACCGT
GLU2_VARIANT_AT2G41220.1.1 GCTGTCCGCGAAAAGAGTGAGACAGCCTATGCAGTTTATCAGCAGCATTTGGCCAACCGT
** ******************************** ************** *********
AT2G41220 CCTATTACGGTTTTCCGCGATCTTCTTGAGTTCAAAAGTGACCGTAACCCTATCCCTGTT
GLU2_VARIANT_AT2G41220.1.1 CCTATTACGGTTTTCCGCGATCTTCTTGAGTTCAAAAGTGACCGTAACCCTATCCCTGTT
************************************************************
AT2G41220 GGGAAGGTTGAGCCTGCTTCCTCTATCGTGGAACGGTTTTGTACAGGTGGAATGTCTCTT
GLU2_VARIANT_AT2G41220.1.1 GGGAAGGTTGAGCCTGCGTCCTCTATTGTGGAACGGTTTTGTACAGGTGGAATGTCTCTT
***************** ******** *********************************
AT2G41220 GGAGCAATCTCAAGAGAAACTCATGAGACAATTGCTATTGCAATGAACAGATTAGGAGGA
GLU2_VARIANT_AT2G41220.1.1 GGAGCAATCTCAAGAGAAACTCATGAGACAATTGCTATTGCAATGAACAGATTAGGAGGA
************************************************************
AT2G41220 AAATCCAACTCAGGGGAAGGAGGCGAGGATCCGATCCGCTGGAAGCCGCTAACAGATGTT
GLU2_VARIANT_AT2G41220.1.1 AAATCCAATTCAGGGGAAGGAGGCGAGGATCCGATCCGCTGGAAGCCGCTAACAGATGTT
******** ***************************************************
AT2G41220 GTTGATGGGTATTCTTCAACTCTGCCACATCTTAAAGGTCTTCGAAATGGGGATACGGCT
GLU2_VARIANT_AT2G41220.1.1 GTTGATGGGTATTCTTCAACACTGCCACATCTTAAAGGTCTTCGAAATGGGGATACGGCT
******************** ***************************************
AT2G41220 ACAAGTGCTATTAAGCAGGTTGCCTCAGGGCGTTTTGGTGTCACTCCGACGTTTTTGGTT
GLU2_VARIANT_AT2G41220.1.1 ACAAGTGCTATCAAGCAGGTTGCTTCAGGGCGTTTTGGTGTCACTCCGACGTTTTTGGTT
*********** *********** ************************************
AT2G41220 AATGCGGACCAGTTGGAAATTAAAGTTGCTCAAGGGGCGAAGCCAGGTGAAGGTGGCCAA
GLU2_VARIANT_AT2G41220.1.1 AATGCGGACCAGTTGGAAATCAAAGTTGCTCAAGGGGCAAAGCCAGGTGAAGGTGGCCAA
******************** ***************** *********************
AT2G41220 CTTCCTGGGAAAAAAGTCAGCGCATATATTGCTAGACTTAGAAATTCTAAACCAGGAGTT
GLU2_VARIANT_AT2G41220.1.1 CTTCCTGGGAAAAAAGTTAGTGCATATATTGCTAGACTTCGAAATTCTAAACCAGGAGTT
***************** ** ****************** ********************
AT2G41220 CCGCTTATTTCTCCACCTCCACATCATGATATTTATTCCATAGAGGATCTGGCACAGTTG
GLU2_VARIANT_AT2G41220.1.1 CCGCTTATTTCTCCACCTCCACATCATGATATTTATTCCATAGAGGATCTGGCACAGTTG
************************************************************
AT2G41220 ATCTTTGATCTCCATCAGGTCAACCCCAAGGCAAAAGTGTCAGTCAAGCTTGTATCAGAA
GLU2_VARIANT_AT2G41220.1.1 ATCTTTGATCTCCATCAGGTCAACCCGAAGGCAAAAGTGTCAGTCAAGCTTGTATCAGAA
************************** *********************************
AT2G41220 ACTGGAATAGGAACAGTTGCCTCCGGAGTAGCGAAGGCTAATGCCGATATCATACAGATA
GLU2_VARIANT_AT2G41220.1.1 GCTGGAATAGGAACAGTTGCCTCGGGAGTAGCGAAGGCTAATGCCGATATCATACAGATA
********************** ************************************
AT2G41220 TCAGGATATGATGGTGGAACTGGAGCCAGCCCTATAAGTTCCATTAAGCATGCTGGTGGC
GLU2_VARIANT_AT2G41220.1.1 TCAGGATATGATGGTGGAACTGGAGCCAGCCCTATAAGTTCCATAAAGCATGCTGGTGGC
******************************************** ***************
AT2G41220 CCGTGGGAGCTGGGACTAGCTGAAACCCAGAAGACTCTCATTGGAAATGGACTAAGGGAA
GLU2_VARIANT_AT2G41220.1.1 CCGTGGGAACTGGGATTAGCTGAAACCCAGAAGACTCTCATTGGAAATGGACTCAGGGAA
******** ****** ************************************* ******
AT2G41220 AGAGTTATTATCAGGGTTGATGGCGGATTCAAAAGTGGTGTGGATGTGTTGATAGCTGCA
GLU2_VARIANT_AT2G41220.1.1 AGAGTTATTATTAGGGTTGATGGCGGATTCAAAAGTGGTGTTGATGTGTTAATAGCTGCA
*********** ***************************** ******** *********
AT2G41220 GCTATGGGCGCTGATGAGTATGGTTTTGGTACTCTGGCAATGATTGCCACAGGATGTATC
GLU2_VARIANT_AT2G41220.1.1 GCTATGGGCGCTGATGAGTATGGGTTTGGTACTCTGGCAATGATTGCCACAGGATGTATC
*********************** ************************************
AT2G41220 ATGGCTCGCATTTGCCACACTAATAACTGCCCCGTTGGTGTTGCCAGTCAGAGAGAGGAA
GLU2_VARIANT_AT2G41220.1.1 ATGGCTCGCATTTGCCACACTAATAACTGCCCTGTTGGTGTTGCCAGTCAGAGAGAGGAA
******************************** ***************************
AT2G41220 CTGCGTGCACGTTTCCCCGGTCTTCCTGGTGATCTTGTCAACTTCTTCTTGTATATTGCA
GLU2_VARIANT_AT2G41220.1.1 CTGCGTGCACGTTTCCCTGGTCTTCCTGGTGATCTTGTCAATTTCTTCTTGTATATTGCA
***************** *********************** ******************
AT2G41220 GAGGAGGTGAGGGGCATATTAGCTCAGCTGGGATATGAAAAGTTGGATGATATAATTGGG
GLU2_VARIANT_AT2G41220.1.1 GAGGAGGTGAGGGGCATATTAGCTCAGTTGGGATATGAAAAACTGGATGATATAATTGGG
*************************** ************* *****************
AT2G41220 CGGACAGATTTGCTAAAGGCTAGGGATATCTCGCTCGTGAAAACTCATCTCGACCTCAGT
GLU2_VARIANT_AT2G41220.1.1 CGGACAGATTTGCTAAAGGCTAGGGATATCTCGCTCGTGAAAACTCATCTTGACCTCAGT
************************************************** *********
AT2G41220 TATCTCTTATCTTCTGTTGGGCTACCAAAACGAAGCAGTACTTCCATTAGGAAGCAGGAG
GLU2_VARIANT_AT2G41220.1.1 TATCTCTTATCTTCTGTTGGGCTACCAAAACGAAGCAGTACTTCTATTAGGAAGCAGGAG
******************************************** ***************
AT2G41220 GTTCATTCAAATGGTCCTGTTCTTGATGATACTCTACTTCAGGATCCAGAGATAATGGAT
GLU2_VARIANT_AT2G41220.1.1 GTTCACTCAAATGGTCCTGTTCTTGATGATACTCTACTTCAGGATCCAGAGATAATGGAT
***** ******************************************************
AT2G41220 GCAATTGAGAATGAAAAGACGGTTCACAAAACCATGAGTATATACAATGTAGATCGTTCT
GLU2_VARIANT_AT2G41220.1.1 GCAATTGAGAATGAAAAAACGGTTCACAAAACCATGAGTATATACAATGTAGATCGTTCT
***************** ******************************************
AT2G41220 GTTTGCGGTCGGATTGCTGGGGTGATTGCGAAGAAATACGGAGACACTGGTTTTGCTGGA
GLU2_VARIANT_AT2G41220.1.1 GTTTGCGGTCGGATTGCTGGGGTGATTGCGAAGAAATACGGAGACACTGGTTTTGCTGGA
************************************************************
AT2G41220 CAACTGAACCTAACGTTCACTGGAAGTGCTGGCCAATCTTTTGCATGTTTCCTAACTCCT
GLU2_VARIANT_AT2G41220.1.1 CAACTGAACCTAACGTTCACCGGAAGTGCTGGCCAATCTTTTGCATGTTTCCTAACCCCT
******************** *********************************** ***
AT2G41220 GGAATGAATATACGTCTTGTGGGTGAAGCCAACGACTATGTGGGAAAGGGAATGGCGGGA
GLU2_VARIANT_AT2G41220.1.1 GGAATGAATATACGTCTTGTGGGTGAAGCCAACGACTATGTGGGAAAGGGAATGGCCGGA
******************************************************** ***
AT2G41220 GGTGAAGTTGTGATATTACCAGTGGAATCGACTGGTTTTCGTCCCGAAGACGCAACTATT
GLU2_VARIANT_AT2G41220.1.1 GGTGAAGTTGTGATATTGCCAGTGGAATCGACTGGTTTTCGTCCCGAAGACGCAACTATT
***************** ******************************************
AT2G41220 GTAGGAAACACTTGTCTGTATGGTGCAACTGGTGGTTTATTATTTGTCAGAGGCAAAGCA
GLU2_VARIANT_AT2G41220.1.1 GTAGGAAACACTTGTCTGTATGGTGCAACTGGTGGTTTATTATTTGTCAGAGGCAAAGCA
************************************************************
AT2G41220 GGGGAGAGATTTGCGGTTAGAAACTCTCTAGCTCAAGCTGTTGTTGAAGGCACTGGAGAT
GLU2_VARIANT_AT2G41220.1.1 GGAGAGAGATTTGCGGTTAGAAACTCTCTAGCTCAAGCTGTTGTTGAAGGCACTGGAGAT
** *********************************************************
AT2G41220 CATTGCTGTGAATATATGACTGGTGGTTGTGTCGTTATACTTGGGAAAGTTGGTAGAAAT
GLU2_VARIANT_AT2G41220.1.1 CATTGCTGTGAATATATGACTGGTGGTTGTGTCGTTGTACTTGGGAAAGTTGGTAGAAAT
************************************ ***********************
AT2G41220 GTAGCCGCTGGGATGACAGGTGGATTGGCATATATCCTTGACGAAGACAACACTCTCCTC
GLU2_VARIANT_AT2G41220.1.1 GTAGCCGCTGGGATGACAGGTGGATTGGCATATATCCTTGACGAAGACAACACTCTCCTC
************************************************************
AT2G41220 CCTAAGATGAACAAAGAGATAGTGAAGATCCAAAGAGTGACTTCACCAGTGGGACAAACA
GLU2_VARIANT_AT2G41220.1.1 CCTAAGATGAACAAAGAGATAGTGAAGATCCAAAGAGTGACTTCACCAGTGGGACAAACA
************************************************************
AT2G41220 CAGCTTAAAAGCCTGATCCAGGCTCATGTGGAGAAAACGGGAAGCAGCAAAGGGGCAATG
GLU2_VARIANT_AT2G41220.1.1 CAGCTTAAGAGCCTGATCCAAGCTCATGTGGAGAAAACGGGAAGCAGCAAAGGAGCAATG
******** *********** ******************************** ******
AT2G41220 ATTGTGGAAGAGTGGGATAAGTATCTAGCAATGTTCTGGCAACTTGTACCTCCGAGTGAA
GLU2_VARIANT_AT2G41220.1.1 ATTGTGGAAGAATGGGATAAGTATCTAGCAATGTTCTGGCAACTTGTACCGCCAAGTGAA
*********** ************************************** ** ******
AT2G41220 GAAGACACACCTGAGGCTAACTCAGACCACATCCTGAAAACAACCACAGGAGATGAAGAA
GLU2_VARIANT_AT2G41220.1.1 GAAGACACACCTGAGGCTAACTCAGACCACATCCTGAAAACAACCACCGGAGATGAAGAA
*********************************************** ************
AT2G41220 CAGGTTTCAAGCACATTAGCAGAGAAGTAA
GLU2_VARIANT_AT2G41220.1.1 CAGGTCTCAAACATATTAGCGGAGAAGTAA
***** **** ** ****** *********

Reviewer 2 Report
The manuscript is the continuation of a previous research and it is interesting as it deals with a very current theme, namely N assimilation under stress conditions. It is well written, although some sentences are very long and on the whole it is not easy to read. Some notes have been included as a comment in the pdf
Author Response
Regulation of Nitrate (NO3) Transporters and Glutamate Synthase Encoding Genes under Drought Stress in Arabidopsis: The Regulatory Role of AtbZIP62 Transcription Factor
Manuscript ID: plants-1405227
Point by point response to the comments of reviewers
We are thankful to the editorial office and anonymous reviewers for their time given to evaluate manuscript. We appreciate their comments, and are happy to share that most of the comments are addressed and have substantially improved the quality of the manuscript. We would like to specify that all changes in the manuscript were highlighted yellow. We hope that the manuscript in the current form will be suitable for publication in the journal.
|
Reviewer 2 |
|
|
The manuscript is the continuation of a previous research and it is interesting as it deals with a very current theme, namely N assimilation under stress conditions. It is well written, although some sentences are very long and on the whole it is not easy to read. Some notes have been included as a comment in the pdf |
We would like to thank the reviewer for his valuable time committed to evaluate this brief report. All the comments given to us have helped us to substantially improve the manuscript. |

Reviewer 3 Report
The article provided by Nkulu Kabange Rolly and Byung-Wook Yun is made at a high methodological level. The authors carried out a thorough analysis of their results. The data are highlighted in sufficient detail. The only thing that causes concern is the low quality of the presentation of materials. The figures in the article are out of order. This is very confusing. In addition, the quality of the drawings is very poor. Figures 2-4 are low resolution and appear blurry. And figure 1 (which is second in number!) Contains very thin lines, and some of the lines are not visible at all. In addition, inside figure 1, the fragments are signed with different font sizes, it looks very casually. It is required to improve the quality of the figures and bring their arrangement in the text according to the content of the article.
I also think that the supplementary section is superfluous. If the authors want to show this information, it can be included in the text of the article, this is not a large table, it will not take up much space. At the present time, this section must either be saturated with additional information or be reduced.
Author Response
Regulation of Nitrate (NO3) Transporters and Glutamate Synthase Encoding Genes under Drought Stress in Arabidopsis: The Regulatory Role of AtbZIP62 Transcription Factor
Manuscript ID: plants-1405227
Point by point response to the comments of reviewers
We are thankful to the editorial office and anonymous reviewers for their time given to evaluate manuscript. We appreciate their comments, and are happy to share that most of the comments are addressed and have substantially improved the quality of the manuscript. We would like to specify that all changes in the manuscript were highlighted yellow. We hope that the manuscript in the current form will be suitable for publication in the journal.
|
Reviewer 3 |
|
|
The article provided by Nkulu Kabange Rolly and Byung-Wook Yun is made at a high methodological level. The authors carried out a thorough analysis of their results. The data are highlighted in sufficient detail. |
We are thankful to the reviewer for his valuable comments and observations that helped us substantially improve the manuscript. We have tried to address the question raised by the reviewer to the best our understanding. |
|
The only thing that causes concern is the low quality of the presentation of materials. The figures in the article are out of order. This is very confusing. In addition, the quality of the drawings is very poor. |
We sincerely apologize for the inconvenience. We have improved the presentation of the figures, their order and citations throughout the manuscript, as suggested. We also apologize for the quality of some of the figures. We have tried to increase the resolution of the Figures. |
|
Figures 2-4 are low resolution and appear blurry. And figure 1 (which is second in number!) Contains very thin lines, and some of the lines are not visible at all. |
We apologize for the inconvenience. Figures with a low resolution where changed with a high resolution of 300 dpi (Figure 2, line 191, page 6, Figure 3, line 249, page 8) |
|
In addition, inside figure 1, the fragments are signed with different font sizes, it looks very casually. It is required to improve the quality of the figures and bring their arrangement in the text according to the content of the article. |
We apologize for the convenience. We have harmonized the font size of the titles of axes in this figure as suggested (Figure 3, line 249, page 8). |
|
I also think that the supplementary section is superfluous. If the authors want to show this information, it can be included in the text of the article, this is not a large table, it will not take up much space. At the present time, this section must either be saturated with additional information or be reduced. |
We appreciate the suggestion made by the reviewer. We have now included the previous Table S1 in the man text as Table 3.
We are not sure of clearly getting the direction of the reviewer’s comment regarding the volume of information. We understood that this comment refers to the supplementary file. If it is the case, the appropriate action has already been taken as indicate above. |
